# Impact of COVID-19 vaccination on preventive behavior: The importance of confounder adjustment in observational studies

**Laura Sità**[1], **Marta Caserotti**[2]*, **Manuel Zamparini**[3], **Lorella Lotto**[2], **Giovanni de Girolamo**[3], **Paolo Girardi**[4]

1 Department of General Psychology, University of Padova, Padova, Italy, 2 Department of Developmental Psychology and Socialization, University of Padova, Padova, Italy, 3 IRCCS Istituto Centro San Giovanni di Dio Fatebenefratelli, Brescia, Italy, 4 Department of Environmental Sciences, Informatics, and Statistics, Ca' Foscari University of Venice, Venezia-Mestre, Italy

* marta.caserotti@unipd.it

## Abstract

The COVID-19 pandemic has underscored the critical role of observational studies in evaluating the effectiveness of public health strategies. However, although many studies have explored the true impact of vaccination on preventive behavior, their results may be skewed by potential biases and confounding variables. This study examines the application of covariate adjustment and propensity score (PS) estimation, particularly through inverse probability treatment weighting (IPTW), to assess their performance in reducing bias in a framework featuring ordinal outcomes and cumulative logistic regression models, as commonly used in observational studies related to social sciences and psychology. Before applying these methods to the case study, we conducted a simulation study that accounted for the presence or absence of model misspecification in an observational scenario with ordinal outcomes, binary treatment, and a continuous confounder. Our findings demonstrate the effectiveness of combining covariate adjustment with PS methods in reducing bias and improving causal inference. These methods were subsequently applied to an Italian observational study on COVID-19 vaccine hesitancy conducted during the initial phase of the vaccination campaign (April-May 2021). Our analysis revealed that vaccination status had a limited short-term impact on the adoption of preventive measures. This study highlights the importance of employing appropriate adjustment techniques in observational research, particularly when evaluating complex behavioral outcomes. The results support the combined use of covariate adjustment and PS methods to enhance the reliability of findings, ultimately contributing to more informed public health decision-making.

## Introduction

During the Coronavirus Disease 2019 (COVID-19) pandemic, the urgent need to develop strategies to contain the spread of the virus led many researchers to conduct observational studies. Indeed, these kinds of studies have played a crucial role in understanding the factors related to the pandemic's spread [1], assessing the effects on people's health [2], and evaluating

**Data Availability Statement:** The data and the R code for reproducing simulations and results are available for download at https://osf.io/68mj5/.

**Funding:** This work was supported by Fondazione Cariplo (grant n° 2020–5195), awarded to Dr. Giovanni de Girolamo, the Italian Ministry of Health (Ricerca Corrente), awarded to Dr. Giovanni de Girolamo, and IRCCS Centro San Giovanni di Dio Fatebenefratelli Institutional resource, awarded to Dr. Giovanni de Girolamo. This work was partially supported by DAIS - Ca' Foscari University of Venice within the IRIDE program, awarded to Paolo Girardi. The funders had no role in study design, data collection and analysis, decision to publish, or preparation of the manuscript.

**Competing interests:** All authors declare that they have no competing interests to be disclosed.

the effectiveness of policies related to restrictive measures [3]. Specifically, the analysis of observational data has been the most commonly employed tool to evaluate the efficacy of public health strategies such as social distancing, mask-wearing, and COVID-19 vaccination campaign. However, the most appropriate study design for estimating the true extent of the treatment effect, according to a causal inference framework, is the randomized controlled trial (RCT), which ensures comparability of outcomes between treated and untreated participants. In particular, the assumption of ignorability [4] guaranteed by the RCT is one of the main necessary conditions for obtaining unbiased results by comparing experimental outcomes between treated and untreated participants [4]. Instead of focusing on individual outcomes, the potential outcome framework [5] allows comparison of a measure related to an outcome of interest between two groups. The Average Treatment Effect (ATE) is a measure used in causal inference to quantify the average effect of a treatment or intervention on an outcome across a population. It represents the average difference in outcomes between the group that receives the treatment and the group that does not as follows

$$ATE = E[Y_1 - Y_o],$$

where $Y_1$ and $Y_o$ are the potential outcomes when the treatment is applied and not applied, while $E[Y_1]$ and $E[Y_0]$ are the averages in case of treatment or no treatment, respectively.

In observational studies estimation of the ATE is likely to be biased because the researcher has little or no control over the independent variable(s), but observes and analyzes data as they naturally occur, due to ethical implications, lack of randomization, and time or cost constraints [6, 7]. In fact, the lack of controlled experimental conditions can alter the observed relationships between treatment and the outcome due to the presence of confounders. Confounders are characteristics that influence both the likelihood of treatment or exposure and the outcome. These include sociodemographic factors such as age, gender, education, and socioeconomic status, which are general confounding variables in observational studies according to the American Psychological Association (APA) guidelines. However, confounders should be carefully selected based on theoretical considerations, including the use of causal diagrams [8].

Implementing adjustment methods in observational studies is crucial for several reasons. First, it helps mitigate confounding, where extraneous variables can skew results. Second, it enhances the accuracy and reliability of findings, thus increasing their generalizability. Third, proper adjustment methods allow researchers to approximate the rigor of randomized controlled trials, thereby strengthening causal inferences. Ultimately, these adjustments are essential for producing valid, credible, and actionable insights from observational data. Covariate adjustment is one of the most popular methods because it allows researchers to quantify effects related to the dependent variable while controlling for a selected set of covariates [9]. However, there are circumstances in which the results of regression models may be affected by issues such as collinearity among covariates/confounders, over-adjustment bias and data sparsity. In such cases, alternative methods should be considered. A useful solution is provided by propensity score (PS) techniques [10], which are defined as the probability of receiving a particular treatment given a set of observed covariates [11]. The estimation and use of propensity scores can be beneficial for balancing covariates, reducing confounding, and handling high-dimensional covariates [12]. This method is particularly advantageous in studies that collect a multitude of potential confounding variables, as it summarizes their effects into a single score [12]. It is also highly effective when the treated and untreated groups have a similar covariate distribution [13], and the confounding effects are limited [14]. Depending on the study design, a common approach involves matching treated (or exposed) and untreated (or unexposed)

individuals to have similar estimates of propensity scores. An alternative approach, called Inverse Probability of Treatment Weighting (IPTW), estimates weights equal to the inverse of the probability of receiving the treatment they received, thereby balancing the distribution of covariates between treated and untreated groups.

In the initial phase of the pandemic, the adopting adjustment procedures was essential to assess, through observational studies, how psychosocial factors, such as risk perception and coping, affected people's adherence to protective behaviors like wearing mask, practicing physical distancing, and frequent hand washing [15, 16]. Similarly, when the vaccine became available, researchers examined how vaccination influenced people's adherence to these behaviors as the pandemic progressed. While vaccination is an effective measure for reducing disease severity and preventing its spread, achieving effective pandemic control requires a comprehensive approach that includes going adherence to preventive measures and addressing vaccine hesitancy. To this end, the aim of this paper is to assess whether vaccination status (received vs. refused) affected adherence to preventive measures against the spread of COVID-19 in Italy during the initial stage of the vaccination campaign (April-May 2021) by using different techniques for confounders adjustment. For this purpose, we analyzed a dataset derived from an Italian observational study on COVID-19 vaccine hesitancy, sponsored by the World Health Organization (WHO) Regional Office for Europe [17]. The structure of this paper is as follows. In the Materials and Methods section, we present the dataset and the adjusting methods used, which include a regression model framework, covariate adjustment, propensity score based on IPTW, and their combination [18]. In the third section, we report the settings and results of a simulation study using a framework commonly employed in social sciences and psychology, which is based on ordinal outcomes and cumulative logistic regression models. The aim of the simulation is to demonstrate how bias can be reduced using two adjustment techniques, and their combination, in two different observational scenarios. In the Results section, we present the results of our study with the scope to estimate the effect of COVID-19 vaccination on the adherence of preventive measures. Finally, the last section offers a commentary on the primary results, with references to the open issues in causal inference and methodological recommendations.

## Materials and methods

### Study setting and participants

The dataset was obtained from the COVID-19 Monitoring run in Italy (COMIT) survey. The survey was conducted in four waves from January to May 2021, and we focused on the third and fourth waves, as the vaccination campaign began during this period in Italy. The questionnaire has been described in detail in previous publications [17, 19]. Out of 5,007 respondents, we selected 1,468 individuals who had been offered the COVID-19 vaccine by local health authorities. Of these, 88.7% reported having received the vaccine. The questionnaire collected various sociodemographic variables (e.g., age group, gender, employment status, education level, and presence of chronic diseases), as well as information on past COVID-19 infection (e.g., yes, no, and don't know). In addition, a panel of validated questions was asked to generate individual factor scores on perceived COVID-19 risk, trust in health institutions, trust in health information sources (e.g., Ministry of Health and WHO), trust in media information sources (e.g., traditional and social media), frequency of use of health information sources, and frequency of use of media information sources [31]. Interviews were conducted by Doxa SPA. Sampling weights were produced comparing the study sample and the Italian population on stratification variables (gender, age, area of residence, size of living centers, education level, and employment status). This study was approved by the Ethics Committee of the

coordinating institution of the national survey (Ethics Committee of the IRCCS San John of God Fatebenefratelli of Brescia, Italy, protocol 286/2020, registration ISRCTN 39724), and all participants provided informed consent. The outcome variables of interest, related to adherence to protective behaviors, were measured by the following questions/statements: 1) I adhered to physical distancing in public (physical distancing); 2) I wore a face mask in public (mask use); 3) I used hand sanitizer when soap and water were not available (hand disinfection). All questions were rated on a 7-point response scale (from 1 = Never to 7 = Always). Fig 1 shows the theoretical relationships in terms of Directed Acyclic Graph (DAG) between outcome (adherence to protective measures), treatment (vaccination status), and a set of confounders.

## Adjustment procedures

In all studies that control for confounders, the analytical strategy used to estimate the causal effect involves the identification, measurement, and appropriate adjustment for a sufficient set

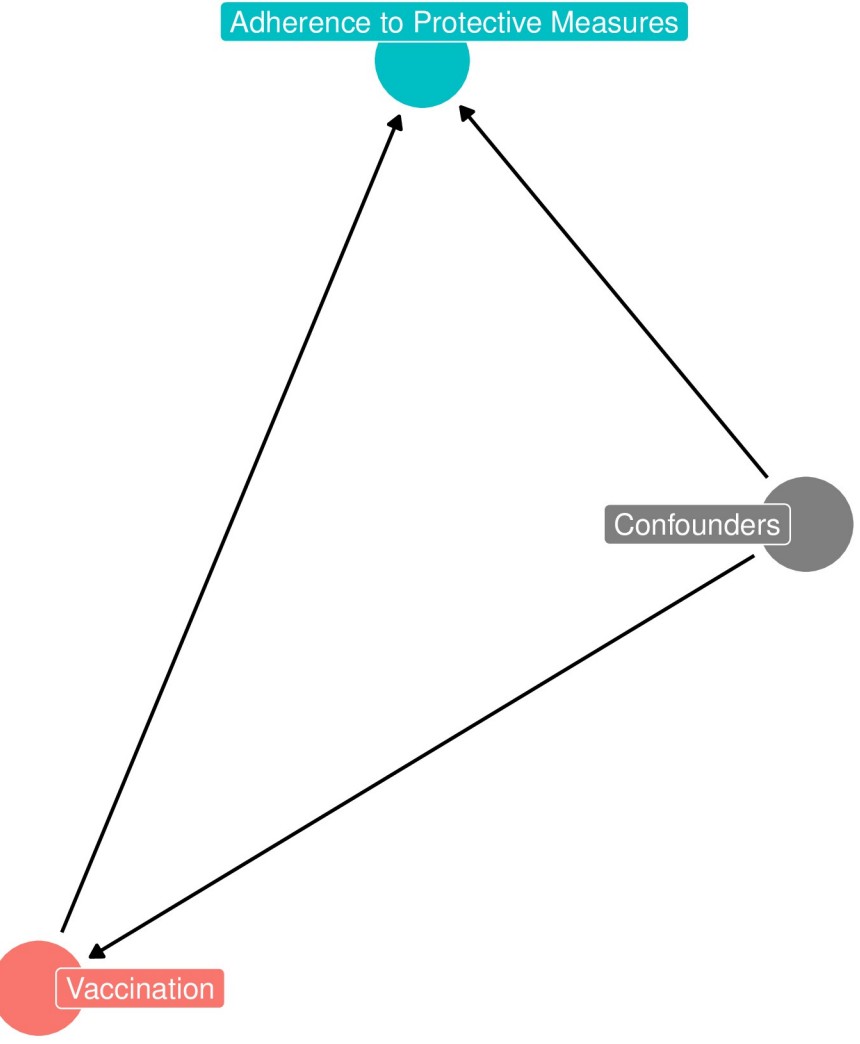

**Fig 1. DAG between adherence to protective measures, vaccination status, and confounders.**

of variables or related proxies [20]. Causal diagrams, including DAGs, can be used to support causal inference, connecting all relevant variables with arrows that represent the relationships, supported by background knowledge, theoretical assumptions, and constraints. Since confounding arises from variables that influence both exposure and outcome, strategies to reduce confounding involve either breaking the association of confounders with outcome (e.g., regression adjustment) or disrupting their association with exposure (e.g., matching, adjustment, or weighting based on PS).

**Covariate adjustment in regression analysis.**   The primary objective is to estimate the impact of the treatment effect on the dependent variable while adjusting for other relevant variables that may affect the results as confounders. The application of multiple linear regression requires the specification of several assumptions, including linearity, independence among observations, and homoscedasticity, to be correctly identified [19]. Although heteroscedasticity can result in an overestimation of the standard error and the significance of the estimates, misrepresenting the model by assuming a linear relationship when it is not can pose a significant issue in covariate adjustment methods. Multiple linear regression with covariate adjustment can be described as follows:

$$Y_i = \alpha_0 + \beta X_i + \gamma_1 Z_{1i} \ldots + \gamma_k Z_{ki} + \epsilon_i,$$

where $Y_i$ is the outcome, $\beta$ is the effect of the treatment variable $X$ (in the simplest case: treatment X = 1, no treatment X = 0), while other coefficients $\gamma_1, \ldots, \gamma_k$ account for the influence of confounders on the outcome $Z_1, \ldots, Z_k$. Finally, the random error term $\epsilon_i$ is generally assumed to be a white noise. In this setting, the estimated value for $\beta$, $\hat{\beta}$, is the ATE estimate. The settings described above can be extended to generalized linear models (GLM). As previously mentioned, Foster noted that although regression has several positive features, its functional form can be misspecified and a few outliers can affect the estimates [21]. In addition, the presence of too many covariates for a given sample size may render the estimate unreliable [22]. One way to potentially address these and other issues is to use alternative methods, such as propensity score (PS) procedures [23].

**Propensity score methods.**   The primary goal of propensity score (PS) is to estimate the probability of receiving or not receiving treatment based on confounders. The usual assumptions in causal framework, such as exchangeability, consistency assumption, and positivity [10, 22], are necessary for the validity of results obtained through PS methods. In addition, the model used to estimate the propensity scores should adequately capture the relationship between the covariates and the treatment assignment, as its misspecification can lead to biased estimates [23]. For an illustration of PS application and a more comprehensive explanation, one may consult the work of Harder and colleagues [24]. PS can be integrated into statistical analysis in multiple ways, the most prevalent of which are matching on PS, stratifying on PS, introducing covariate adjustment using PS, and calculating IPTW using PS [25]. For the simulation and the real data application, we selected IPTW, as weighting each individual (treated and control) allows us to consider the full sample population without losing any subjects (e.g., through PS stratification). All statistical analyses and simulations were performed using R Statistical Program, version 4.3 [26], with the *Weightit* R package. All the simulation code, statistical analyses, figures, and tables can be reproduced using the materials available in the online OSF repository https://osf.io/68mj5/.

## Regression models

Given the ordinal nature of the response variables, we employed weighted cumulative logistic models (CLM) assuming proportional odds and using the sampling weights provided by the

survey company. For the model with covariate adjustment we selected a set of variables that included sociodemographic characteristics (e.g., age, gender, education level, and employment status), past COVID-19 contagion, and standardized factor scores related to COVID-19 perceived risk, conspiracy, trust in health institutions, trust in health information sources, trust in media information sources, frequency of use of health information sources, and frequency of use of media information sources. More details on the estimation of the factor scores can be found in the study by de Girolamo et al. [17]. The inclusion of the latter set of variables is essential, as they generally affect both adherence to protective behaviors and COVID-19 vaccination. In addition we included in the PS and CLM models a series of interaction terms between gender and some key variables (age, educational level, COVID-19 perceived risk, and conspiracy) [27, 28] The results of the CLM regression models are presented in terms of Odds Ratios (ORs). A 95% Wald confidence interval (CI) was used for ORs estimated with the covariate adjustment, while for the PS method, we accounted for the uncertainty of weights creation by calculating the relative 95% CI using 500 bootstrap replications.

## Simulations

### The theoretical model

The proposed simulation aims to serve as an illustrative example of the application of the two previously described approaches, covariate adjustment, PS weighting, and their combination, in the context of bias reduction. We considered the same classical theoretical model, represented by the DAG in Fig 1, in which the treatment exerts a direct effect on the outcome, while we considered for simplicity the presence of only one confounder (i.e., age) affecting both the outcome and the treatment.

In behavioral and social science, it is common to consider measurements based on a Likert scale as dependent variables, as reported in the dataset under study. Accordingly, we considered a generative cumulative logistic regression model based on the assumption of proportional odds. In a first simulative scenario, the generative model is denoted as follows

$$g(P(Y_i \leq j)) = \alpha_j + \beta X_i + \gamma Z_i,$$

for $i = 1, \ldots n$, where $n$ is the sample size, $\alpha_j$ is an intercept that varies with j = 2,..., J = 5 with $Y_i$ is an ordinal outcome with 5 categories, $X$ is a binary predictor denoting the presence of a treatment (1 = treatment; 0 = no treatment), while the function $g()$ is a logit link function. The variable $Z_i$ is a confounder while the values $\beta$ and $\gamma$ regulate the strength of the treatment and of the covariate-to-outcome confounding, respectively.

In this formulation, $X$ is expressed as a binary variable. The confounder variable $Z_i$ is assumed to follow a uniform distribution between 0 and 1. The values related to the coefficients $\alpha_j$ are set to have a symmetric baseline probability of 0.15, 0.20, 0.30, 0.20, and 0.15, for each category from 1 to 5, respectively.

To include a covariate-to-treatment confounding effect, the term $Z_i$ modifies the probability of an individual being under-treatment as follows

$$X_i \sim Bernoulli(\tau_i = Z_i),$$

in which the individual probability of treatment assignment $\tau_i$ varies with $Z_i$. Since $Z_i$ is distributed according to a uniform random variable ($Z_i \sim U(0,1)$), the average probability to be assigned to the treatment is 50%.

In addition to this first scenario in which we generated and estimated the coefficients with same model, we considered a second scenario modifying the generative cumulative model

$$g(P(Y_i \leq j)) = \alpha_j + \beta X_i + \gamma Z_i^2,$$

and the relationship between the treatment and confounders

$$X_i \sim Bernoulli\left(\tau_i = Z_i^2\right),$$

keeping fixed all the other conditions. Compared to the first simulation settings, the second scenario proposed unfavorable conditions for procedures based on covariate adjustment since the estimating model is misspecified.

## Simulation settings and bias estimation

In our simulation, for each scenario we varied the size of the sample ($N$ = 200, 500, 1,000, and 2,000), while for the effect size we accounted for low, medium, and high effects ($\beta$ = 0.25, 0.5, and 1.0 that result in Odds Ratio = exp ($\beta$) = 1.24, 1.65, and 2.72, respectively), varying also the strength of the confounders to treatment parameter ($\gamma$ = 1 and 2). Considering the conditions of the first scenario with $\gamma$ = 1 and $\beta$ = 0.5, an example of the relationship among these three variables is shown in Fig 2 in which the pairwise comparison among X, Y, and Z is made by graphical representations (boxplot, histogram, bar plot, and mosaic plot). A discrete correlation was observed between the confounder (Z) and the outcome (Y) (Spearman's correlation ~ 0.23) with higher values of the continuous confounder at increasing of the outcome category. The distribution of the confounder variable visually differs considering the treated (X = 1) and untreated group (X = 0). Lastly, the distribution of the outcome (Y) reports higher frequency for modalities 4–5 for treated individuals compared to non-treated ones. The data were simulated using the function *genOrdCat* of the package *simstudy* under R environment.

Four different approaches were used to estimate $\beta$: method A) cumulative logistic regression model without covariate (age) adjustment ($Y_i = \beta X_i$); method B) cumulative logistic regression model with covariate (age) adjustment ($Y_i = \beta X_i + \gamma Z_i$); method C) cumulative logistic regression model PS weights without covariate (age) adjustment; and method D) cumulative logistic regression model with PS weights and covariate (age) adjustment. The cumulative regression models of methods A and B were estimated using the function *clm* of the R package *ordinal*, while for models C and D we used the function *ordinal_weightit* of the R package *Weighit*.

The first step of a PS weighting method can be achieved using a logistic regression model in which treatment status (treatment $X_i$ = 1, or control $X_i$ = 0) is regressed on the confounding covariates. In recent years, the literature has reported a wide range of procedures spacing from adaptive binary regression, machine learning, and predictive models with strategies of variable selection and regularization to predict the propensity scores. Adopting a simple and flexible approach, we estimated each individual weight ($P(X_i = 1) = \hat{\pi}_i$, and $P(X_i = 0) = 1 - \hat{\pi}_i$) using a Generalized Boosted Model (GBM) which provides more precise estimates than classical logistic model, especially when the relationship between X and Z is complex and non-linear. GBM is based on decision trees and creates a predictive procedure by combining multiple simple models using an iterative algorithm. We set the number of trees to 200 and a learning rate of 0.1, since the relationship between X and Z was simple and to speedup simulations. For more details about the choice of GBM parameters see Elith et al. [29]. Then, the stabilized weights (WS) were adopted to improve precision of the causal effect estimates, and they were

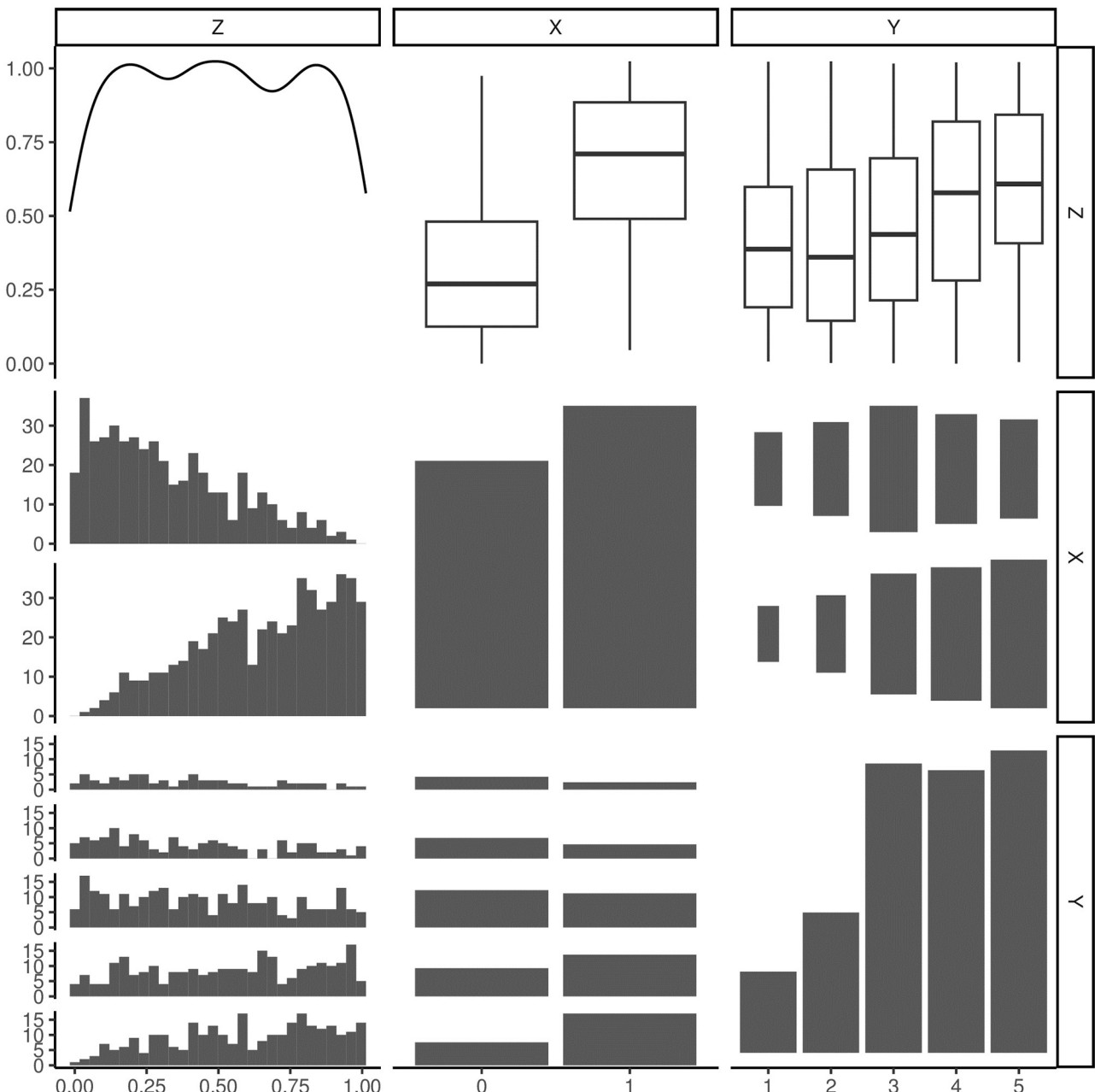

**Fig 2. Bivariate and marginal distributions of a simulated dataset of the first scenario with $n = 1000$, $\beta = 0.5$, and $\gamma = 1$.**

calculated as

$$WS_i = \frac{p_x}{\hat{\pi}_i} \text{ for } X = 1, \text{ and } WS_i = \frac{1 - p_x}{1 - \hat{\pi}_i} \text{ for } X = 0,$$

where $p_x$ is the proportion of treated individuals ($p_x = (\sum_{i=1}^n X_i)/n$).

The estimation bias of $\beta$ is evaluated using the Average Relative bias (ARebias) as follows

$$\text{ARebias} = \frac{1}{K}\sum_{k=1}^{K}\frac{(\hat{\beta}_k - \beta_{true})}{\beta_{true}},$$

where the $\hat{\beta}_k$ is the estimation of the k-th simulation. As a measure of variability of the estimates we also calculated the Mean Squared Error (MSE) as follows

$$\text{MSE} = \frac{1}{K}\sum_{k=1}^{K}\left(\hat{\beta}_k - \beta_{true}\right)^2.$$

We performed K = 1000 simulations for each combination of $\beta$, $\gamma$, and $n$.

## Simulation results

As reported in Table 1, when considering the simulation settings of the first scenario, method A (no adjustment) a positive and high degree of bias was found, with a higher average relative bias (ARebias) observed for higher values of γ. A decrease in ARebias values was observed with increasing effect size β (Table 1). Bias was consistently reduced with all adjustment procedures. The reduction in bias with effect size and sample size was generally monotonic, while an increase was observed when the confounder on the outcome parameter (γ) moved from 1 to 2. The performance in terms of ARebias of method D (PS weighting with covariate) was comparable to that of method B (covariate adjustment), while only PS showed a slightly worse performance. For large sample sizes (N = 2000), the bias was always less than 5%. Taking into account the variability of the estimates, method B showed the lowest MSE compared to the other adjustment methods. It is crucial to contextualize the interpretation of the results within the specific simulation setting in which the model was estimated, considering the same form of the generative one. This first scenario created a favorable condition for effective bias reduction by covariate adjustment. The results changed when the second scenario was considered. As shown in Table 2, the presence of model misspecification implied a strong increase in the relative bias for all methods. Method A (no adjustment) showed a strong and positive bias, which was higher when the confounder-to-treatment coefficients were equal to 2 (compared to γ = 1); a decrease in ARebias values was observed with increasing sample size and the treatment to outcome coefficient β (Table 2). Bias was partially reduced with all adjustment procedures. Bias reduction for covariate adjustment (method B) was only partially achieved, with ARebias values above 5% in most cases. ARebias values were also higher for models adjusted by PS weighting (method C), but generally only for smaller effect sizes (β ≤ 1). Method D (PS weighting with covariate) obtained the best performance in terms of bias reduction (≤5%), except for smaller effect and sample sizes. As reported in the previous scenario, the MSE was lower for method B than for the other three methods.

## Results

The main characteristics of participants by vaccination status are shown in Table 3: COVID-19 vaccinated individuals had higher levels of education (Standardized Mean Difference, SMD = 0.300) and were more likely to work in the health sector (SMD = 0.272) compared to those who refused vaccination. The difference in age and gender by vaccination status was limited (SMD = 0.033 and 0.158, respectively). Compared to those who refused the vaccine, vaccinated individuals were more likely to have no history of COVID-19 infection (SMD = 0.366) and to report no chronic diseases (SMD = 0.210). Respondents who received the vaccine reported higher scores for all protective measures compared to those who were not vaccinated,

**Table 1. First scenario.** ARebias (standard error) and MSE by combination of effect size ($\beta$), strength of the confounder-to-outcome parameter ($\gamma$) and population size ($n$) for the model without (A) and with adjustment method (B = covariate, C = PS, D = PS+covariate). In bold |ARebias| higher $\geq$ 5%.

| n | $\beta$ | $\gamma$ | ARebias | | | | MSE | | | |
|---|---|---|---|---|---|---|---|---|---|---|
| | | | A | B | C | D | A | B | C | D |
| 200 | 0.25 | 1 | **1.318 (0.103)** | 0.005 (0.127) | **0.146 (0.144)** | 0.040 (0.154) | 0.175 | 0.100 | 0.130 | 0.149 |
| 500 | 0.25 | 1 | **1.319 (0.066)** | -0.031 (0.080) | 0.026 (0.093) | **-0.050 (0.098)** | 0.136 | 0.040 | 0.054 | 0.060 |
| 1000 | 0.25 | 1 | **1.304 (0.045)** | 0.004 (0.055) | 0.043 (0.064) | -0.006 (0.066) | 0.119 | 0.019 | 0.026 | 0.027 |
| 2000 | 0.25 | 1 | **1.306 (0.033)** | -0.010 (0.040) | 0.033 (0.046) | -0.004 (0.048) | 0.113 | 0.010 | 0.014 | 0.014 |
| 200 | 0.5 | 1 | **0.672 (0.054)** | 0.022 (0.065) | **0.070 (0.074)** | 0.033 (0.078) | 0.185 | 0.105 | 0.139 | 0.151 |
| 500 | 0.5 | 1 | **0.655 (0.034)** | 0.012 (0.040) | 0.047 (0.045) | 0.022 (0.046) | 0.136 | 0.040 | 0.051 | 0.053 |
| 1000 | 0.5 | 1 | **0.658 (0.025)** | 0.011 (0.029) | 0.029 (0.034) | 0.012 (0.035) | 0.124 | 0.022 | 0.029 | 0.030 |
| 2000 | 0.5 | 1 | **0.652 (0.016)** | 0.003 (0.020) | 0.022 (0.022) | 0.013 (0.023) | 0.113 | 0.010 | 0.012 | 0.013 |
| 200 | 1 | 1 | **0.325 (0.027)** | 0.001 (0.032) | 0.026 (0.039) | 0.019 (0.040) | 0.177 | 0.104 | 0.151 | 0.162 |
| 500 | 1 | 1 | **0.329 (0.017)** | 0.007 (0.020) | 0.013 (0.024) | 0.012 (0.024) | 0.137 | 0.040 | 0.057 | 0.058 |
| 1000 | 1 | 1 | **0.324 (0.012)** | 0.005 (0.014) | 0.006 (0.017) | 0.007 (0.017) | 0.119 | 0.020 | 0.027 | 0.028 |
| 2000 | 1 | 1 | **0.326 (0.009)** | 0.006 (0.010) | 0.001 (0.012) | 0.005 (0.012) | 0.114 | 0.010 | 0.014 | 0.014 |
| 200 | 0.25 | 2 | **2.625 (0.106)** | **0.083 (0.132)** | **0.252 (0.141)** | **0.083 (0.154)** | 0.501 | 0.110 | 0.129 | 0.148 |
| 500 | 0.25 | 2 | **2.538 (0.067)** | -0.019 (0.082) | **0.085 (0.090)** | -0.013 (0.097) | 0.430 | 0.042 | 0.051 | 0.059 |
| 1000 | 0.25 | 2 | **2.529 (0.046)** | 0.010 (0.056) | **0.066 (0.062)** | 0.000 (0.066) | 0.413 | 0.019 | 0.024 | 0.027 |
| 2000 | 0.25 | 2 | **2.521 (0.033)** | 0.005 (0.040) | 0.046 (0.043) | 0.012 (0.046) | 0.404 | 0.010 | 0.012 | 0.013 |
| 200 | 0.5 | 2 | **1.255 (0.054)** | -0.001 (0.066) | **0.070 (0.072)** | 0.019 (0.078) | 0.468 | 0.108 | 0.132 | 0.152 |
| 500 | 0.5 | 2 | **1.266 (0.034)** | 0.026 (0.049) | **0.050 (0.045)** | 0.033 (0.048) | 0.430 | 0.042 | 0.052 | 0.057 |
| 1000 | 0.5 | 2 | **1.237 (0.024)** | -0.008 (0.029) | -0.005 (0.032) | -0.003 (0.033) | 0.397 | 0.020 | 0.025 | 0.028 |
| 2000 | 0.5 | 2 | **1.255 (0.017)** | 0.018 (0.021) | 0.001 (0.023) | 0.017 (0.024) | 0.401 | 0.011 | 0.013 | 0.014 |
| 200 | 1 | 2 | **0.606 (0.029)** | 0.005 (0.034) | 0.000 (0.039) | 0.012 (0.040) | 0.451 | 0.116 | 0.150 | 0.163 |
| 500 | 1 | 2 | **0.609 (0.018)** | 0.014 (0.021) | -0.005 (0.025) | 0.019 (0.025) | 0.404 | 0.046 | 0.061 | 0.064 |
| 1000 | 1 | 2 | **0.595 (0.013)** | -0.006 (0.015) | -0.035 (0.017) | -0.001 (0.017) | 0.370 | 0.021 | 0.029 | 0.029 |
| 2000 | 1 | 2 | **0.593 (0.009)** | -0.003 (0.010) | -0.044 (0.012) | -0.003 (0.012) | 0.360 | 0.010 | 0.016 | 0.014 |

suggesting slightly higher compliance with these measures among vaccinated individuals. In addition, respondents who refused the vaccine perceived less risk from COVID-19, reported higher conspiracy scores, lower trust in health institutions and information sources (both health and media), and less use of media information (all SMDs > 0.3). To avoid numerical instability of the estimating algorithm due to low frequencies on the first 2 categories (1 and 2), we recoded the outcome variables (physical distancing, mask use, and hand disinfection) to be a 5-point Likert scale collapsing the first two categories into the third one.

Due to the observational nature of this study, we cannot directly assess the effect of vaccination status on adherence to protective behaviors because differences in covariates (e.g., perceived COVID-19 risk, conspiracy, etc...) could have influenced both the variables of interest and vaccination status. To compare the effect of vaccination on adherence to protective behaviors, we used covariate and PS weighting adjustments. Since we are interested specifically in the effect of the vaccination on those who received it, we estimated the PS weights in order to estimate the Average Treatment effect of the Treated (ATT); at this end, we set the scores for vaccinated individuals to 1, estimating the probability of refusal among those who declined the vaccine [30].

Given the PS model, we estimated the probability of inclusion using Generalized Boosted Model based on the same set of covariates used for covariate adjustment and weighted by sampling weights. We adopted the default options for GBM parameters (number of trees:10000;

**Table 2. Second scenario.** ARebias (standard error) and MSE by combination of effect size ($\beta$), strength of the confounder-to-outcome parameter ($\gamma$) and population size ($n$) for the model without (A) and with adjustment method (B = covariate, C = PS, D = PS+covariate). In bold |ARebias| higher $\geq$ 5%.

| n | β | γ | ARebias | | | | MSE | | | |
|---|---|---|---|---|---|---|---|---|---|---|
| | | | A | B | C | D | A | B | C | D |
| 200 | 0.25 | 1 | **1.568 (0.110)** | **0.157 (0.141)** | **0.151 (0.182)** | **-0.116 (0.202)** | 0.229 | 0.126 | 0.207 | 0.256 |
| 500 | 0.25 | 1 | **1.615 (0.073)** | **0.170 (0.090)** | **0.147 (0.115)** | **-0.068 (0.125)** | 0.196 | 0.052 | 0.085 | 0.098 |
| 1000 | 0.25 | 1 | **1.569 (0.049)** | **0.163 (0.061)** | **0.143 (0.082)** | -0.032 (0.089) | 0.169 | 0.025 | 0.044 | 0.049 |
| 2000 | 0.25 | 1 | **1.582 (0.035)** | **0.174 (0.043)** | **0.087 (0.061)** | **-0.063 (0.065)** | 0.164 | 0.013 | 0.023 | 0.027 |
| 200 | 0.5 | 1 | **0.799 (0.056)** | **0.094 (0.069)** | **0.108 (0.095)** | -0.003 (0.103) | 0.238 | 0.120 | 0.227 | 0.265 |
| 500 | 0.5 | 1 | **0.787 (0.036)** | **0.093 (0.043)** | **0.095 (0.059)** | 0.005 (0.064) | 0.186 | 0.049 | 0.090 | 0.101 |
| 1000 | 0.5 | 1 | **0.787 (0.026)** | **0.088 (0.031)** | **0.065 (0.041)** | -0.014 (0.044) | 0.171 | 0.025 | 0.043 | 0.048 |
| 2000 | 0.5 | 1 | **0.779 (0.017)** | **0.075 (0.022)** | 0.043 (0.031) | -0.026 (0.033) | 0.159 | 0.014 | 0.025 | 0.027 |
| 200 | 1 | 1 | **0.402 (0.029)** | **0.061 (0.036)** | **0.072 (0.048)** | 0.029 (0.051) | 0.246 | 0.133 | 0.237 | 0.265 |
| 500 | 1 | 1 | **0.396 (0.018)** | 0.049 (0.023) | 0.032 (0.031) | -0.005 (0.033) | 0.191 | 0.056 | 0.098 | 0.110 |
| 1000 | 1 | 1 | **0.387 (0.013)** | 0.042 (0.016) | 0.024 (0.023) | -0.005 (0.024) | 0.165 | 0.026 | 0.054 | 0.058 |
| 2000 | 1 | 1 | **0.386 (0.009)** | 0.037 (0.011) | 0.004 (0.015) | -0.022 (0.016) | 0.157 | 0.013 | 0.023 | 0.026 |
| 200 | 0.25 | 2 | **3.126 (0.115)** | **0.384 (0.142)** | **0.387 (0.178)** | **-0.065 (0.198)** | 0.694 | 0.135 | 0.207 | 0.245 |
| 500 | 0.25 | 2 | **3.057 (0.071)** | **0.321 (0.088)** | **0.256 (0.114)** | **-0.115 (0.126)** | 0.616 | 0.055 | 0.085 | 0.100 |
| 1000 | 0.25 | 2 | **3.032 (0.051)** | **0.313 (0.062)** | **0.206 (0.083)** | **-0.107 (0.091)** | 0.591 | 0.030 | 0.045 | 0.053 |
| 2000 | 0.25 | 2 | **3.015 (0.036)** | **0.293 (0.045)** | **0.136 (0.059)** | **-0.136 (0.065)** | 0.576 | 0.018 | 0.023 | 0.028 |
| 200 | 0.5 | 2 | **1.510 (0.057)** | **0.161 (0.068)** | **0.177 (0.091)** | -0.032 (0.102) | 0.652 | 0.123 | 0.213 | 0.257 |
| 500 | 0.5 | 2 | **1.525 (0.036)** | **0.186 (0.044)** | **0.118 (0.059)** | -0.032 (0.064) | 0.613 | 0.056 | 0.089 | 0.103 |
| 1000 | 0.5 | 2 | **1.494 (0.026)** | **0.158 (0.032)** | **0.069 (0.043)** | **-0.056 (0.047)** | 0.574 | 0.031 | 0.047 | 0.056 |
| 2000 | 0.5 | 2 | **1.492 (0.018)** | **0.149 (0.021)** | 0.044 (0.030) | **-0.059 (0.033)** | 0.564 | 0.017 | 0.023 | 0.028 |
| 200 | 1 | 2 | **0.726 (0.031)** | **0.073 (0.036)** | 0.042 (0.049) | -0.024 (0.053) | 0.623 | 0.135 | 0.244 | 0.278 |
| 500 | 1 | 2 | **0.729 (0.019)** | **0.084 (0.022)** | 0.036 (0.032) | -0.011 (0.033) | 0.569 | 0.057 | 0.101 | 0.110 |
| 1000 | 1 | 2 | **0.730 (0.014)** | **0.083 (0.016)** | 0.014 (0.023) | -0.018 (0.024) | 0.552 | 0.033 | 0.052 | 0.058 |
| 2000 | 1 | 2 | **0.719 (0.010)** | **0.073 (0.012)** | -0.007 (0.017) | -0.026 (0.018) | 0.527 | 0.020 | 0.028 | 0.032 |

learning rate: 0.01 for binary) since the model complexity was high due to the presence of several covariates. The loveplot diagram shown in Fig 3 suggests an acceptable covariate balancing provided by PS procedure, with almost all SMDs below 0.2 and most of them below 0.1. After applying propensity score weighting, among who refused the vaccine the effective sample size (ESS) was reduced from the original sample size of 159.8 to 69.8. This ESS reduction is primarily due to the evident differences between the treated and control groups in terms of baseline covariates. While this reduction implies a decrease in the precision of our estimates, the weighting procedure is essential to correct for potential confounding and to ensure that the treated and untreated groups are comparable.

The resulting distribution of weights is shown in S1 Fig, indicating a wide range of weights distribution for the untreated group (vaccination refused). Model results in terms of OR and 95% CI are reported in Table 4.

Without any adjustment (method A), adherence to physical distancing increased among the vaccinated respondents (vs. refused; OR = 1.45, 95% CI = 1.07–1.96). However, all adjusted models showed no effect of vaccination status on this protective measure (OR $\approx$ 1). A similar result was obtained for adherence to mask use, where the unadjusted model showed a statistically significant increase in this behavior among those who chose to be vaccinated (OR = 1.64; 95% CI = 1.20–2.23). In contrast, all adjusted models estimated an OR close to 1, but PS with covariate adjustment method suggested a decrease in adherence to mask use (OR ~ 0.75),

**Table 3. Main characteristics of respondents by vaccination status.**

| Variable | Vaccination status | | SMD[2] |
|---|---|---|---|
| | Refused, N = 166[1] | Received, N = 1,302[1] | |
| **Gender** [Females] | 85 (51.2%) | 645 (49.5%) | 0.033 |
| **Age class** | | | 0.158 |
| 18–34 | 22 (13.3%) | 187 (14.4%) | |
| 35–44 | 27 (16.3%) | 163 (12.5%) | |
| 45–54 | 49 (29.5%) | 336 (25.8%) | |
| 55–70 | 68 (41.0%) | 616 (47.3%) | |
| **Educational level** (years of study) | | | 0.300 |
| 0–8 | 71 (42.8%) | 394 (30.3%) | |
| 9–13 | 64 (38.6%) | 532 (40.9%) | |
| 13+ | 31 (18.7%) | 376 (28.9%) | |
| **Employment status** | | | 0.272 |
| Unemployed | 70 (42.2%) | 589 (45.2%) | |
| Employed—Not healthcare system | 89 (53.6%) | 577 (44.3%) | |
| Employed–Healthcare system | 7 (4.2%) | 136 (10.4%) | |
| **Past COVID-19 contagion** | | | 0.366 |
| No | 123 (74.1%) | 1113 (85.5%) | |
| I don't know | 24 (14.5%) | 55 (4.2%) | |
| Yes | 19 (11.4%) | 134 (10.3%) | |
| **Chronic disease presence** | | | 0.210 |
| No | 98 (59.0%) | 839 (64.4%) | |
| I don't know | 14 (8.4%) | 47 (3.6%) | |
| Yes | 54 (32.5%) | 416 (32.0%) | |
| **Physical distancing** | | | 0.361 |
| 1 | 11 (6.6%) | 29 (2.2%) | |
| 2 | 7 (4.2%) | 33 (2.5%) | |
| 3 | 13 (7.8%) | 39 (3.0%) | |
| 4 | 18 (11%) | 161 (12%) | |
| 5 | 39 (23%) | 293 (23%) | |
| 6 | 29 (17%) | 324 (25%) | |
| 7 | 49 (30%) | 423 (32%) | |
| **Mask use** | | | 0.351 |
| 1 | 1 (0.6%) | 1 (<0.1%) | |
| 2 | 4 (2.4%) | 3 (0.2%) | |
| 3 | 3 (1.8%) | 20 (1.5%) | |
| 4 | 12 (7.2%) | 40 (3.1%) | |
| 5 | 24 (14%) | 174 (13%) | |
| 6 | 44 (27%) | 284 (22%) | |
| 7 | 78 (47%) | 780 (60%) | |
| **Hand disinfection** | | | 0.539 |
| 1 | 13 (7.8%) | 8 (0.6%) | |
| 2 | 11 (6.6%) | 13 (1.0%) | |
| 3 | 3 (1.8%) | 28 (2.2%) | |
| 4 | 14 (8.4%) | 69 (5.3%) | |
| 5 | 23 (14%) | 166 (13%) | |
| 6 | 33 (20%) | 278 (21%) | |

*(Continued)*

**Table 3.** (Continued)

| Variable | Vaccination status | | SMD[2] |
|---|---|---|---|
| | Refused, N = 166[1] | Received, N = 1,302[1] | |
| 7 | 69 (42%) | 740 (57%) | |
| **COVID-19 perceived risk** | -0.21 (1.00) | 0.07 (0.75) | 0.318 |
| **Conspiracy** | 0.38 (0.85) | -0.10 (0.88) | 0.554 |
| **Trust in health institutions** | -0.57 (1.07) | 0.18 (0.85) | 0.778 |
| **Trust in health information sources** | -0.67 (1.14) | 0.17 (0.84) | 0.849 |
| **Trust in media information sources** | -0.34 (1.07) | 0.08 (0.87) | 0.431 |
| **Frequency of use of health information source** | -0.54 (0.98) | 0.21 (0.90) | 0.795 |
| **Frequency of use of media information sources** | -0.26 (0.84) | 0.04 (0.87) | 0.345 |

[1]Mean (SD) or frequency (%).

[2]Standardized Mean Difference.

although the result was not statistically significant. A different result was reported for the adherence to hand disinfection: the unadjusted model showed a strong positive association with vaccination status (received vs. refused; OR = 2.21, 95% CI = 1.63–3.01), while the models with covariate and PS adjustment showed a weak positive, though not statistically significant, association between vaccination and hand disinfection.

## Discussion

The present study demonstrates the practical value of the adjustment techniques in assessing the influence of COVID-19 vaccine uptake on COVID-19-related behaviors. We analyzed a subset of respondents who were offered the vaccine at the beginning of the Italian vaccination campaign. The results illustrate how vaccination status and the inclusion of adjustment procedures can provide more reliable evidence on the impact of vaccination on key preventive measures against COVID-19. We also evaluated the adjustment capabilities of the two most commonly used confounder adjustment techniques, namely covariate adjustment, propensity score, as well as their combination, through a simulation study, considering a typical framework in observational studies related to social sciences and psychology, where the outcome takes an ordinal form (i.e., Likert scale). In our simulation study, the use of a propensity score procedure with inverse probability weighting resulted in a slightly higher relative bias compared to covariate adjustment, especially when the confounding variable had a strong effect on the outcome. When the model was misspecified, IPTW methods showed a reduced degree of bias compared to covariate adjustment, especially when IPTW was combined with covariate adjustment.

The PS methods have become increasingly popular for adjusting for confounding in observational studies [25, 31]. The results of our simulation study refer to two specific scenarios. In the first scenario, the estimation model was correctly specified. In this context, no significant differences were found between the three adjustment methods, although covariate adjustment showed some superiority in terms of reduction and variability of the estimate. The superiority of covariate adjustment over PS has previously been demonstrated in the context of logistic regression models [32], particularly in the presence of strong confounder-to-outcome effects [33]. In the second scenario, we introduced model misspecification for both the confounder-to-outcome and the treatment-to-outcome relationships. All methods showed some degree of

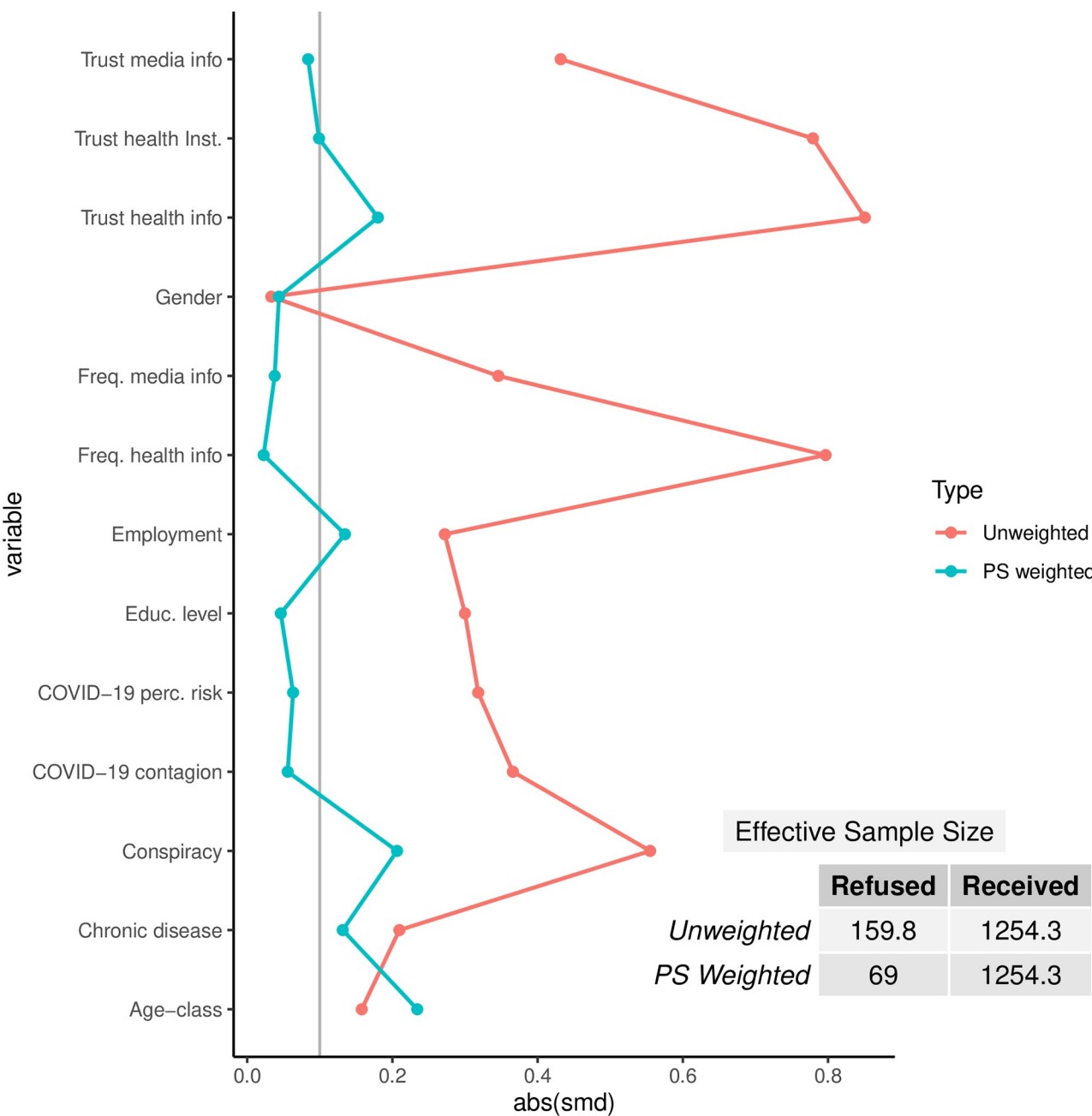

**Fig 3. Loveplot diagram.** Estimated standardized mean difference (smd) and effective sample size (ESS) before and after PS weighting procedure.

relative bias, with PS methods demonstrating superior performance due to the flexibility of GBM-based techniques [34]. However, simulations considering a larger number of scenarios and adjustment techniques have shown that each adjustment method has its own advantages and disadvantages, and no single method is consistently superior [5, 35]. Furthermore, in the absence of a strong theoretical model, model misspecification cannot be excluded. Our simulation showed that using PS weighting with covariate adjustment can be an effective solution, as reported by Nguyen et al [18] and other related proposals [34]. A method that performs well

**Table 4. Non-adjusted (A) and adjusted ORs (B, C, and D) for social distancing, mask use, and hand disinfection by vaccination status (done vs. refused) for the four models (B = covariate adjustment, C = PS, D = PS with covariate adjustment).**

| | A | B | C | D |
|---|---|---|---|---|
| | OR (95% CI) | OR (95% CI) | OR (95% CI) | OR (95% CI) |
| **Physical distancing** | | | | |
| Vaccination status [received] | 1.45* (1.07–1.96) | 0.96 (0.69–1.32) | 0.95 (0.60–1.51) | 0.96 (0.60–1.54) |
| **Mask use** | | | | |
| Vaccination status [received] | 1.64* (1.20–2.23) | 0.96 (0.68–1.36) | 0.81 (0.51–1.26) | 0.74 (0.47–1.18) |
| **Hand disinfection** | | | | |
| Vaccination status [received] | 2.21* (1.63–3.01) | 1.39 (0.99–1.95) | 1.24 (0.81–1.90) | 1.20 (0.78–1.85) |

*p-value<0.05.

under model misspecification is more desirable than one that is optimized for an ideal scenario [36]. In addition, PS expands the ability to examine outcomes in observational studies across a wide range of scenarios, partially addressing confounding effects and improving the internal validity of study results [37].

The utility and relevance of considering different confounder adjustment techniques are also evident when the literature shows mixed results. On one side, Yamamura et al. [38] using panel data collected from March 2020 to September 2021 (N = 54,007) found that vaccinated individuals were more likely to stay home, wash their hands frequently and wear masks than unvaccinated individuals, consistently from the early stages of COVID-19 until after vaccine distribution. Continuing data collection until September 2022 (N = 70,908) comparing preventive behaviors before and after vaccination, the same working group [39] showed that after vaccination people increased preventive behaviors, such as mask use and hand hygiene. These authors also showed that this behaviour was predicted by pro-social motivation. On the other side, several recent studies have observed that COVID-19 vaccination leads to a decrease in participants' compliance with public health measures [40, 41]. Another study that surveyed participants before the first dose and after their second dose reported a decreased willingness to use a face mask and to maintain physical distancing [42]. However, these authors observed that hand hygiene was preferred over physical distancing [42]. In addition, other studies conducted in Italy reported a decline over time in behaviors or attitudes related to compliance with protective behaviors toward COVID-19 [43, 44]. A large observational study (12 countries and 80,305 respondents) reported no differences in physical distancing between those who had received one vaccine dose and those who had not, but differences were observed between the latter and those who had completed the vaccination cycle [45]. Interestingly, using PS adjustment, Wright et al. [46] reported no change in adherence to protective measures between October 2020 and March 2021 after COVID-19 vaccination, although a gradual decline in overall adherence was observed from the first year of the pandemic.

In our study, unadjusted analyses showed that those who chose to be vaccinated were more likely to engage in protective behaviors compared to those who did not. In contrast, adjusted methods showed that vaccinated individuals did not change their short-term adherence to preventive measures, with an apparent reduction in mask use in the PS models, though not statistically significant due to large confidence intervals. The results are consistent with recent evidence: COVID-19 vaccination had a limited effect on adherence to protective behaviors. The possible reduction in adherence to protective behaviors may be attributed to the perceived lower probability of infection due to vaccine protection. Most

people tend to be selective in how they evaluate data in everyday life; they are more likely to accept explanatory models that reduce risk and increase perceived benefit in their own experience [47]. Another possible interpretation of the results can be related to the moral hazard theory [48], according to which people underestimate or ignore the collective consequences of their actions, assuming that the efforts of others (in this case others adopting protective behaviours) will continue to protect them, resulting, however, in public health, economic and social costs to the community [49].

It is important to emphasize that adjustment methods are particularly relevant in applied psychology research: causal inference is an essential tool for designing interventions for target populations, as it represents a unique method for identifying interventions a priori [21]. Failure to identify underlying assumptions or relying on implicit and opaque inferences exacerbates the replicability crisis in psychological science [50]. This issue can hinder the growth of cumulative research and may lead researchers or readers to mistakenly interpret ambiguous findings as causal [51].

Finally, in the specific context of the considered study, we emphasize that several assumptions must be verified in order to derive causal inference from observational studies using adjustment methods. Regarding PS methods, the plausibility of the exchangeability assumption can be strengthened by carefully selecting covariates, particularly those related to trust in health institutions, health information sources, and conspiracy beliefs, as these are key drivers of vaccine hesitancy and adherence to preventive measures [52]. Consistency can be ensured through the extensive efforts made during the COVID-19 vaccination campaign to monitor and verify vaccine administration. Finally, it is crucial to consider the potential violation of the no-interference assumption, especially in the context of vaccination and adherence to protective behaviors [53]. In particular, the vaccination of an individual may result in the phenomenon of "free riding," whereby those who have refused the vaccine may still benefit from the protection provided by vaccinated individuals due to concerns about potential herd immunity. However, the study was conducted during the early phase of the vaccination campaign and we can assume that the interference should be limited. Another crucial point concerns the COVID-19 vaccination campaign: in the initial phase, vaccination was primarily offered to vulnerable populations (e.g., elderly, immunosuppressed, with comorbidities, etc...) and to essential workers (e.g., health workers, civil servants, etc...). The specific selection of the population invited for vaccination may limit the external validity of the results and, consequently, the generalizability of the findings, despite the incorporation of sampling weights. Lastly, the WHO dataset was based on self-reported questions, which may raise concerns about item validity.

## Conclusions

This study convincingly demonstrates the critical importance of adjustment techniques in observational research, particularly when evaluating the impact of COVID-19 vaccination on preventive behaviors. Analysis of data from the initial Italian vaccination campaign shows that the uptake of the vaccine uptake had only a marginal effect on adherence to protective behaviors, with a slight decrease in mask use. This is likely attributable to a perceived reduction in the risk of infection due to the vaccine's protective properties. It is important to note that this study demonstrated the significant enhancement in results reliability that can be achieved through the use of covariate adjustment, propensity score (PS) methods and their combination. In scenarios of model misspecification, PS methods demonstrated superior performance, consistent with existing research, highlighting their flexibility in handling confounders.

## Supporting information

**S1 Fig. Distribution of the estimated stabilized weights by vaccination status.**
(TIF)

## Author Contributions

**Conceptualization:** Laura Sità, Marta Caserotti, Paolo Girardi.

**Data curation:** Paolo Girardi.

**Formal analysis:** Paolo Girardi.

**Funding acquisition:** Lorella Lotto, Giovanni de Girolamo.

**Investigation:** Laura Sità, Marta Caserotti, Paolo Girardi.

**Methodology:** Laura Sità, Paolo Girardi.

**Project administration:** Lorella Lotto.

**Software:** Laura Sità, Paolo Girardi.

**Supervision:** Lorella Lotto, Giovanni de Girolamo.

**Validation:** Laura Sità, Marta Caserotti, Giovanni de Girolamo, Paolo Girardi.

**Visualization:** Marta Caserotti, Manuel Zamparini, Lorella Lotto, Giovanni de Girolamo, Paolo Girardi.

**Writing – original draft:** Laura Sità, Marta Caserotti, Paolo Girardi.

**Writing – review & editing:** Laura Sità, Marta Caserotti, Manuel Zamparini, Lorella Lotto, Giovanni de Girolamo, Paolo Girardi.

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
