## [Decision Letter · Decision Letter 0]

17 Mar 2024

PONE-D-24-03437Biased Findings without Adjusting Methods in Observational Studies: Can Vaccination Status Influence Uptake of Preventive Measures against COVID-19?PLOS ONE

Dear Dr. Girardi,

Thank you for submitting your manuscript to PLOS ONE. After careful consideration, we feel that it has merit but does not fully meet PLOS ONE’s publication criteria as it currently stands. Therefore, we invite you to submit a revised version of the manuscript that addresses the points raised during the review process.

Your manuscript requires a major revision.You will have to address in details all the comments and execute changes and suggestions made by both reviewers in order for the manuscript to be accepted.For details, please see the reviewers comments.

We look forward to receiving your revised manuscript.

Kind regards,

Srebrenka Letina, Ph.D.

Academic Editor

PLOS ONE

“This work was supported by Fondazione Cariplo (grant n° 2020-5195), the Italian Ministry of Health (Ricerca Corrente), and IRCCS Centro San Giovanni di Dio Fatebenefratelli institutional resources.”

3. In this instance it seems there may be acceptable restrictions in place that prevent the public sharing of your minimal data. However, in line with our goal of ensuring long-term data availability to all interested researchers, PLOS’ Data Policy states that authors cannot be the sole named individuals responsible for ensuring data access (http://journals.plos.org/plosone/s/data-availability#loc-acceptable-data-sharing-methods).

Additional Editor Comments:

Major revision required.

Reviewers' comments:

Reviewer's Responses to Questions

**Comments to the Author**

1. Is the manuscript technically sound, and do the data support the conclusions?

Reviewer #1: Partly

Reviewer #2: Partly

2. Has the statistical analysis been performed appropriately and rigorously? 

Reviewer #1: No

Reviewer #2: No

3. Have the authors made all data underlying the findings in their manuscript fully available?

Reviewer #1: No

Reviewer #2: No

4. Is the manuscript presented in an intelligible fashion and written in standard English?

Reviewer #1: Yes

Reviewer #2: Yes

5. Review Comments to the Author

Reviewer #1: The manuscript compares the bias of the ANCOVA method, and three versions of inverse probability of treatment weighted (IPTW) methods for the estimation of the average causal effect of a point-treatment on an outcome, and demonstrates all four methods using an empirical example concerning the effect of COVID-19 vaccinations on adherence to preventive measures. Personally, I am very much in favor of promoting methods that can adjust for large sets of covariates, as confounding seems to be a major is issue in (psychological, and social science) research that aims to answer causal questions (sometimes implicitly so) and using nonexperimental data. In that sense I support the author’s overarching goal with this manuscript. At the same time, after having finished reading the manuscript, I do have a major concern, which I will elaborate on below. Following, I will elaborate on other concerns/questions that I had. I hope the authors find these comments useful for the improvement of the manuscript.

My major concern is that I was unsure about the “novelty” or “added-value” of the manuscript in relation to the vast amount of literature that already exists about this topic. As a didactical paper about PS-based methods, the manuscript falls short as it skips over many of the important details. While the authors do refer to other literature for the interested reader, such as Schafer and Kang (2008), I believe the manuscript does not offer any new insightful way of explaining the methods compared to the vast introductory literature that already exists and that goes into great detail (cf. Hernán and Robins, 2021; Naimi, Cole and Kennedy, 2016; Vansteelandt and Sjolander, 2016). As a methodological paper, the simulations are too simplified and limiting to provide any new insight into the performance of these methods in particular scenarios. The conclusion that unadjusted methods lead to bias in the presence of confounding, and PS-based methods (if the PS-model is correctly specified) not, is not new, and neither is the conclusion that IPTW-based methods can be incredibly inefficient (Robins, Hernán, Brumback, 2001). As an applied paper, the manuscript also falls short of many of the considerations and decisions researchers must make in practice (e.g., evaluation of post-balancing imbalance, evaluation of plausibility of conditional exchangeability and consistency assumptions). For improving the manuscript, I would advice the authors to make more explicit what this paper contributes to the existing literature.

A second concern is the treatment of selection bias in the manuscript. It is mentioned multiple times in the manuscript (e.g., lines 78, 134), and it plays a role in the simulation study (see Section 3.2). However, it is unclear to me what point the researchers are trying to convey with this, as the primary focus in the introduction, discussion and conclusion is about confounding bias.

Some (relatively) minor comments are mentioned below:

1. In line 55, there is mention of “non-probability samples”. How does this term relate to the rest of the introduction, in which the focus is predominantly on confounding bias? Is “non-probability sample” a means of saying that the treatment is not randomized, or do the authors refer to selection bias here?

2. In line 57, the authors mention that “The use of regression models is preferred for large samples because (…)”. Is this true *generally*? Schafer and Kang (2008) state “[PS-based] methods are in part a reaction to misapplications of ANCOVA, where analysts were often unaware of the sensitivity of their results to model failure.”. At the least, such a statement seems unnuanced, and there are definitely situations with large samples in which PS-based methods are preferred.

3. Line 70 mentions that use of propensity scores can improve precision. This seems unnuanced, as IPTW-based methods are known to be inefficient (Vansteelandt and Joffe, 2014).

4. The last two sentences of this paragraph (lines 71-74) seem rather unrelated to the introduction of multiple PS-based methods, and read a bit weirdly. Personally, I would remove them, as these techniques from econometrics play no further role in this manuscript.

5. Line 77 is the first line that introduces IPTW. Given that this plays a major role in the simulations, I would introduce it earlier, rather than mentioning PS-methods generally.

6. Lines 102-103, this sentence is unclear to me.

7. Lines 108-111 mentions parametric assumptions for the ANCOVA method to work well. However, this seems nuanced again: You can also specify nonlinear outcome models with interactions. When this model is correct, then the coefficient of the binary exposure can be interpreted as a causal effect estimate. Furthermore, homoscedasticity is, I believe, not an issue for the bias of point estimates, but only for unbiased estimation of the standard errors (but please check me on this).

8. Lines 123-127 mentions causal identification assumptions for the PS-methods. However, these methods must equally hold for ANCOVA-methods (with ANCOVA, you also need no unmeasured confounders, for example). Similarly, the parametric assumptions mentioned for the ANCOVA method equally apply for PS-based methods, but now the PS-model needs to be correctly specified.

9. Line 134, Stabilized weights are not used because they reduce selection bias, right? Rather, because they reduce the variance of the weights and hopefully also improve precision of the causal effect estimates.

10. The Equation on line 167 includes Cohen’s d. What is the added value of using Cohen’s d here rather than simply picking some reasonable population value?

11. What is the reasoning for using B-splines, especially given the simple linear relationships in the generated data? I understand that in practice you don’t know if the true relationships are indeed linear, but the use of B-splines comes a little of the blue here.

12. For the results in Tables 1 and 2, what are the Monte Carlo SE’s of the bias estimates here (see Morris et al., 2016, for definition of this performance measure)? Because, in these results, the estimated bias for the scenario with n = 4000 and an effect of 2 (0.2 * d), the bias of the PS-based method is still approximately -0.2. This seems far off, given that both the outcome model and the PS-model should be able to rightly capture the linear relationships in the generated data.

13. In line 228, it is mentioned that “(…) with the estimated bias increasing with effect size (…)”. Indeed, a larger effect means more variance in the outcome, and logically speaking also a larger bias (right?). Therefore, wouldn’t it be better to report relative bias, rather than absolute bias (see Morris et al., 2016).

14. In line 235 you interpret estimates for gamma. Why would you want to interpret this parameter if your research question is only about the effect of X on Y? I think that most causal inference literature urges researcher not to interpret the parameters of confounders as these might be biased themselves, and the inclusion of these covariates only serves to adjust for confounding for the specific target parameter.

15. In Table 3, why use p-values? If you are assessing covariate balance, usually metrics like standardized mean differences (SMDs) are reported. Or if you are just describing sample characteristics, than p-values are not needed because you are not attempting to generalize to the population (i.e., there is no uncertainty about the characteristics of the people in the sample, as everyone in the sample was actually observed).

16. In Table 3, the outcome measures are also included. Are these actually end-of-study outcomes, or are these outcomes measured as earlier waves (and thus serve as confounders for the relationship between treatment and end-of-study outcome).

17. Line 293, please evaluate if weighting was successful by assessing, for example, SMDs in the balanced sample.

18. Line 296, how are these CI’s computed? Based on robust SEs, bootstrap, or regular SEs?

19. Line 331, you mention “For a subset of respondents who reported being vaccinated at the beginning of (…)”. This isn’t the target population, right? It is those that were invited. Also, those that were invited might be a particular subpopulation of most vulnerable people. How does this influence the external validity of the results?

20. Empirical example generally, a serious treatment of the empirical example also warrants a serious evaluation of the causal identification assumptions that underlie the methods employed here. This is currently lacking. Do the authors think that the assumptions of exchangeability, consistency, positivity, and no interference (seems important in the context of vaccines) are plausible?

Reviewer #2: This paper contributes to further understanding of the use of adjusting methods in observational studies. However, the key concern is that the distribution for the outcome used for simulation study is different from the one used for observational data. This makes the findings obtained from simulation studies less relevant to the observation study. Therefore, I recommend author to undertake simulation studies where the distribution of outcome variable is the same as the one used for observational studies. They also need to consider a situation when those assumptions are not met and evaluate how the results change. Considering this is a simulation study authors should share their analysis codes or links to enable study reproducibility, and this will help when paper is being reviewed.

I therefore recommend, major revision for the manuscript.

6. PLOS authors have the option to publish the peer review history of their article (what does this mean?). If published, this will include your full peer review and any attached files.

Reviewer #1: No

Reviewer #2: No

---

## [Author Response · Author response to Decision Letter 0]

27 May 2024

Dear Editor and Reviewers,

The revised version of the manuscripts now includes all the format changes suggested by the editor and the additional information required. All comments and suggestions have been addressed in the separately uploaded Word file.

Best regards

---

## [Decision Letter · Decision Letter 1]

22 Jul 2024

PONE-D-24-03437R1Biased Findings without Adjusting Methods in Observational Studies: Can Vaccination Status Influence Uptake of Preventive Measures against COVID-19?PLOS ONE

Dear Dr. Girardi,

Thank you for submitting your manuscript to PLOS ONE. After careful consideration, we feel that it has merit but does not fully meet PLOS ONE’s publication criteria as it currently stands. Therefore, we invite you to submit a revised version of the manuscript that addresses the points raised during the review process.

In agreement with reviewer 1, I think the manuscript needs to address in detail their comments in its next revision in order to be accepted.  I agree that the intended contribution(s) of the manuscript are unclear. I suggest focusing on one clear contribution and making it very strong instead on trying to tackle three different types of contributions and not being convincing enough in neither. This will require rewriting some parts of your paper as well as possibly changing its current title, because in it is current form I think it is misleading as it suggest a review or a didactical paper about the specific methodology. You are asked to addressed all reviewer 1's comments in detail. 

We look forward to receiving your revised manuscript.

Kind regards,

Srebrenka Letina, Ph.D.

Academic Editor

PLOS ONE

Reviewers' comments:

Reviewer's Responses to Questions

**Comments to the Author**

1. If the authors have adequately addressed your comments raised in a previous round of review and you feel that this manuscript is now acceptable for publication, you may indicate that here to bypass the “Comments to the Author” section, enter your conflict of interest statement in the “Confidential to Editor” section, and submit your "Accept" recommendation.

Reviewer #1: (No Response)

Reviewer #2: All comments have been addressed

2. Is the manuscript technically sound, and do the data support the conclusions?

Reviewer #1: No

Reviewer #2: Yes

3. Has the statistical analysis been performed appropriately and rigorously? 

Reviewer #1: No

Reviewer #2: Yes

4. Have the authors made all data underlying the findings in their manuscript fully available?

Reviewer #1: No

Reviewer #2: No

5. Is the manuscript presented in an intelligible fashion and written in standard English?

Reviewer #1: No

Reviewer #2: Yes

6. Review Comments to the Author

Reviewer #1: Review for PLOS ONE, manuscript PONE-D-24-03437_R1

Biased Findings without Adjusting Methods in Observational Studies: Can Vaccination Status Influence Uptake of Preventive Measures against COVID-19?

Thank you for the ability to review this revised manuscript. The authors have clearly put a lot of effort into incorporating the feedback, and in parts, the manuscript sees some big improvements. At the same time, these changes have, in my opinion, not adequately addressed my major concern of the previous review, which is that the added value of this manuscript in relation to the vast (introductory) literature on causal inference methods is unclear to me. I will elaborate on this below by commenting on this manuscript’s statistical contribution to the literature, didactical contribution to the literature, and substantive contribution to the literature. I end with a list of further critical comments that I wrote up while reading the manuscript. I hope the authors find these comments useful for the improvement of their manuscript.

First, as a statistical contribution, the simulations are simply too limited, and focus on scenarios that have been previously investigated by other researchers as well. For example, in line 156 it is written: “A more comprehensive evaluation of different adjustment strategies in different simulation settings can be found in Schafer and Kang [28].”. Actually, Schafer and Kang (2008) did also include inverse probability weighting in their simulations, and this, in addition to the other simulation studies into IPW estimation that are references in the reference list of the manuscript, makes one wonder what the added-value is of these simulations? This is not clear in the current manuscript. Furthermore, the topic of double adjustment (i.e., as in Method D) has also already been extensively studied in Nguyen et al. (2017, doi: 10.1186/s12874-017-0338-0).

Second, as a didactical contribution, the manuscript falls short. My main concern here is that there are many different terms being used, but that the authors do not clearly describe how these are all related? I discuss some of these issues below in my list of further comments, but just a few examples: The authors mention the ignorability assumption, but do not explain that this (strong ignorability) is actually the combination of the exchangeability and positivity; the authors also mention assessing overlap in covariates, but do not explain that it is related to assessing the positivity assumptions; and the authors mention in line 116 the assumption of “independence” (I assume the authors refer to independence of residuals in a statistical model here), but then do not explain how this relates to exchangeability. In summary, the presentation of the assumptions that these methods rely on is simply unstructured, and therefore unclear for novices to IPW-based methods. It also again begs the question what is new that this manuscript brings to the table, especially compared to didactical introductions to IPW-estimation by the likes of Noah Greifer (https://ngreifer.github.io/WeightIt/articles/estimating-effects.html) and Naimi, Cole, and Kennedy (2017).

Third, as a substantive contribution to the literature, the manuscript also makes too many shortcuts (i.e., many important details are unexplained). For example:

- Are the sampling weights still valid when only a selection of the sample was used?;

- Why does the L.O.V.E. plot in Table S1 not all the age classes, educational levels, employment statuses, etc. (shouldn’t these categorical covariates have dummies, which are used for weighting?);

- Why were standard Wald confidence intervals used when it is recommended to use boostrapping to take the uncertainty of creating the weights into account (see Greifer, 2024, at https://ngreifer.github.io/WeightIt/articles/estimating-effects.html#estimating-standard-errors-and-confidence-intervals).

- Why was only a subsect of covariates selected for adjustment, as implied on line 318?

All in all, it is unclear to me what this manuscript is supposed to be. The further comments below also list some statements in the manuscript which I believe to be false (or unnuanced), and concerns regarding the validity of the simulation study results. All of these comments combined make me conclude that the manuscript is not ready for publication yet.

Further comments:

- In general, there are many spelling mistakes and grammatically awkward sentences. For example, frequently the word “causal” is written as “casual”, and Figure 3 contains the word “counfounder” rather than “confounder”.

- L. 45. “(…) the assumption of ignorability guarantees the unbiasedness of results (…)”. While this is a necessary assumption, unbiasedness is not guaranteed per se. If you have a continuous exposure and wrongly assume the exposure-outcome relationship is linear, then you can still have biased results (due to model misspecification) even when ignorability holds.

- L 53. Shouldn’t it be “(…) Y1 and Y0 are the potential outcomes (…)”?

- L. 55. “This does not apply to observational studies.” Disagree, it does apply also in observational studies, as this is just the definition of the effect of interest. However, you cannot simply take the means of both groups to estimate both effects, so the way to identify and estimate the estimand is different, but the estimand can still be defined as the ATE.

- L 57. “preferred” compared to what? Furthermore, alternative modeling approaches, like IPW-estimation, can also "(…) quantify effects related to the dependent variable while controlling for a limited set of covariates.", so why is this characteristic of regression models an advantage per se?

- L. 61. Not true, as the presence of multiple confounders is not necessarily a problem for outcome regression methods. Furthermore, model misspecification can also occur in IPW methods (when your exposure model is misspecified), so I don’t understand the point being made here.

- L. 68. Does the target audience of this paper know what is meant by "overlapping distribution of covariates"?

- L. 70-76. First, the authors introduce propensity scores. Then, in lines 70 to 76, the authors talk about stratification based on individual covariates. This seems like a weird shift? Wouldn’t you stratify based on the propensity score? Furthermore, in the lines 70 to 73, it is mentioned that we need to match for confounders, and then in line 74 it is said that we need to additionally consider “(…) characteristics that influence both the probability of the exposure and outcome (…)”. How are these characteristics in line 74 different from the confounders in lines 70 to 73?

- Line 76-79. The sentence is unclear to me. What is also unclear is how reference 16 is relevant, which is about extending results from an RCT to a wider target population of interest, to stratification that is discussed earlier in this paragraph.

- Line 81. Although IPW estimation is the main focus of the paper, it is only mentioned in line 81. This is confusing as matching is discussed earlier in the introduction, but actually not at all what the paper focusses on.

- L. 110. What is a “likely” impact?

- L. 112. Confounders are not just “thought to be responsible”, the ARE actually responsible for difference in characteristics between treated and untreated. Furthermore, should a confounder also impact the outcome, for it to be a confounder? This is missing from this sentence.

- L. 114. What does “specification of several assumptions to be correctly identified” mean?

- L. 116. Why is “no interaction with the treatment variable” a key assumption? As long as I include an interaction when there is one, then I can still estimate my ATE. Admittedly, the coefficient itself is not the ATE anymore, but you can still estimate the ATE with g-computation, which involves multiple regression analysis.

- L. 136. Here three fundamental assumptions are discussed, but aren’t these the assumptions of exchangeability and positivity, discussed 1 sentence earlier?

- Figure 2. It needs further explanation, because I have no idea what I am looking at.

- L. 217. The statement that GBM provided more precises estimates seems unnuanced. Perhaps for the scenario with a quadratic effect of Z, but not for the scenario with only a linear effect. Also, are you worried about overfitting when using GBMs? Is that an issue?

- L. 228. Given that IPTW-based method can be quite inefficient, it seems appropriate to also look at a measure of variability of the estimates. For example, the MSE?

- L. 231. I do not see how this statement is backed up by the results. Comparing n = 200 to n = 500, relative bias is lower for n = 500. However, then it increases again. Furthermore, what is the uncertainty around these ARebias estimates? I think they overlap, as the point estimates of ARebias are so close. In that case, you wouldn’t conclude that ARebias decreased with increasing sample size.

- L. 234. Why is there a heightened risk of bias in all cases, as all relationships have been correctly specified, right? (i.e., they are all linear)

- L. 248. Why is there still bias for method C? Should GBM be able to handle the quadratic relationship between Z and exposure? If so, then balancing should work, right?

- L. 266. For the empirical example, wouldn’t you say you are more interested in the average treatment effect in the treated (the ATT) rather than the average treatment effect (ATE), generally?

- L. 312. "(...) we can't directly assess (...)" or "(...) we can't asses the direct effect (...)". In case of the latter, that would be a weird statement, given that this manuscript is all about trying to estimate causal effects by using adjustment methods.

- L. 409. Can you ever say the assumption of exchangeability was met? Usually, exchangeability is presented as an untestable assumption, which can simply be made more or less plausible through thorough discussion of the included (and not included) covariates.

Reviewer #2: Dear Editor,

Authors have adequately addressed both the major and minor comments. Overall, this study will contribute to the current literature (both methodological and substantively) about covariate adjustment and propensity score methods. They have also clearly demonstrated both the strength and limitations of the approaches they have used under different scenarios which was missing in the first draft. They have shared the analysis code though I would suggest they make it open access where someone does not need to request permission from authors to access. For example, as part of supplementary material.

Therefore, having read the revised paper and response to reviewer comments response I recommend the paper to be accepted in the current form.

7. PLOS authors have the option to publish the peer review history of their article (what does this mean?). If published, this will include your full peer review and any attached files.

Reviewer #1: No

Reviewer #2: No

---

## [Author Response · Author response to Decision Letter 1]

16 Sep 2024

Editor

Thank you for submitting your manuscript to PLOS ONE. After careful consideration, we feel that it has merit but does not fully meet PLOS ONE’s publication criteria as it currently stands. Therefore, we invite you to submit a revised version of the manuscript that addresses the points raised during the review process.

In agreement with reviewer 1, I think the manuscript needs to address in detail their comments in its next revision in order to be accepted. I agree that the intended contribution(s) of the manuscript are unclear. I suggest focusing on one clear contribution and making it very strong instead on trying to tackle three different types of contributions and not being convincing enough in neither. This will require rewriting some parts of your paper as well as possibly changing its current title, because in it is current form I think it is misleading as it suggest a review or a didactical paper about the specific methodology. 

You are asked to addressed all reviewer 1's comments in detail.

Authors: Thank you for your constructive feedback. We recognize the importance of clarifying the manuscript's contribution and have refocused it to emphasize a single, clear contribution. While our initial approach aimed to address multiple contributions, we agree that this diluted the overall impact. Consequently, we have revised the manuscript to present a substantial contribution to the literature on the impact of COVID-19 vaccination on behavioral preventive measures, specifically through the lens of various adjustment methods. Additionally, the title has been revised to better reflect the content. We have also thoroughly addressed all of Reviewer 1's comments in detail within our review. 

Reviewer #1: Review for PLOS ONE, manuscript PONE-D-24-03437_R1

Biased Findings without Adjusting Methods in Observational Studies: Can Vaccination Status Influence Uptake of Preventive Measures against COVID-19?

Thank you for the ability to review this revised manuscript. The authors have clearly put a lot of effort into incorporating the feedback, and in parts, the manuscript sees some big improvements. At the same time, these changes have, in my opinion, not adequately addressed my major concern of the previous review, which is that the added value of this manuscript in relation to the vast (introductory) literature on causal inference methods is unclear to me. I will elaborate on this below by commenting on this manuscript’s statistical contribution to the literature, didactical contribution to the literature, and substantive contribution to the literature. I end with a list of further critical comments that I wrote up while reading the manuscript. I hope the authors find these comments useful for the improvement of their manuscript.

First, as a statistical contribution, the simulations are simply too limited, and focus on scenarios that have been previously investigated by other researchers as well. For example, in line 156 it is written: “A more comprehensive evaluation of different adjustment strategies in different simulation settings can be found in Schafer and Kang [28].”. Actually, Schafer and Kang (2008) did also include inverse probability weighting in their simulations, and this, in addition to the other simulation studies into IPW estimation that are references in the reference list of the manuscript, makes one wonder what the added-value is of these simulations? This is not clear in the current manuscript. Furthermore, the topic of double adjustment (i.e., as in Method D) has also already been extensively studied in Nguyen et al. (2017, doi: 10.1186/s12874-017-0338-0).

Second, as a didactical contribution, the manuscript falls short. My main concern here is that there are many different terms being used, but that the authors do not clearly describe how these are all related? I discuss some of these issues below in my list of further comments, but just a few examples: The authors mention the ignorability assumption, but do not explain that this (strong ignorability) is actually the combination of the exchangeability and positivity; the authors also mention assessing overlap in covariates, but do not explain that it is related to assessing the positivity assumptions; and the authors mention in line 116 the assumption of “independence” (I assume the authors refer to independence of residuals in a statistical model here), but then do not explain how this relates to exchangeability. In summary, the presentation of the assumptions that these methods rely on is simply unstructured, and therefore unclear for novices to IPW-based methods. It also again begs the question what is new that this manuscript brings to the table, especially compared to didactical introductions to IPW-estimation by the likes of Noah Greifer (https://ngreifer.github.io/WeightIt/articles/estimating-effects.html) and Naimi, Cole, and Kennedy (2017).

Third, as a substantive contribution to the literature, the manuscript also makes too many shortcuts (i.e., many important details are unexplained). For example:

- Are the sampling weights still valid when only a selection of the sample was used?;

- Why does the L.O.V.E. plot in Table S1 not all the age classes, educational levels, employment statuses, etc. (shouldn’t these categorical covariates have dummies, which are used for weighting?);

- Why were standard Wald confidence intervals used when it is recommended to use boostrapping to take the uncertainty of creating the weights into account (see Greifer, 2024, at https://ngreifer.github.io/WeightIt/articles/estimating-effects.html#estimating-standard-errors-and-confidence-intervals).

- Why was only a subsect of covariates selected for adjustment, as implied on line 318?

All in all, it is unclear to me what this manuscript is supposed to be. The further comments below also list some statements in the manuscript which I believe to be false (or unnuanced), and concerns regarding the validity of the simulation study results. All of these comments combined make me conclude that the manuscript is not ready for publication yet.

Authors: Thank you for your detailed review and insightful comments. We appreciate the opportunity to improve our manuscript based on your feedback. We believe that the third option, making a substantive contribution to the literature, is our primary goal. While we recognize that our initial submission raised concerns regarding clarity and value, we hope the revised manuscript more effectively highlights the novel aspects and significance of our work Specifically, we have clarified that the added value of our simulations and methodological parts is to support the interpretation of the results related to COVID-19 vaccine data. In particular, we have provided a more detailed explanation of our decisions in relation to the substantive contribution to the literature.

- The sampling weights remain valid even if some of the data have been selected, as our focus has been on ensuring internal validity. Achieving external validity in this context is challenging because the relative selection of the population is generally influenced by a number of factors, such as pandemic dynamics, government policies, and population characteristics, that may vary in each specific observational study. 

- To calculate the Standardized Mean Differences (SMD) for categorical variables, we employed the method proposed by Yang and Dalton (2012) (Yang, D., & Dalton, J. E. (2012, April). A unified approach to measuring the effect size between two groups using SAS. In SAS global forum (Vol. 335, pp. 1-6)).

- We chose to use Wald confidence intervals solely for covariate adjustment, while we used bootstrap confidence intervals for propensity score methods.

- We selected a subset of covariates for adjustment based on the original dataset, which contained a large amount of information not all relevant to this study. We focused on variables that were potential confounders in relation to the behavioral outcomes and the vaccination status.

To better reflect the change in the nature of the paper, we have also changed the title of the manuscript. In addition, we have strengthened the methodological part according to your comments regarding the statistical and didactic contributions. All sections have been thoroughly revised according to your comments.

Further comments:

- In general, there are many spelling mistakes and grammatically awkward sentences. For example, frequently the word “causal” is written as “casual”, and Figure 3 contains the word “counfounder” rather than “confounder”.

Authors: We have carefully reviewed the document with the help of a native speaker. We hope to have fully addressed the reported errors and grammatical issues.

- L. 45. “(…) the assumption of ignorability guarantees the unbiasedness of results (…)”. While this is a necessary assumption, unbiasedness is not guaranteed per se. If you have a continuous exposure and wrongly assume the exposure-outcome relationship is linear, then you can still have biased results (due to model misspecification) even when ignorability holds.

Authors: We have rephrased the sentence to clarify the concept. In particular, in an RCT we can compare the outcome distribution without a regression model (so misspecification is not an issue), just by calculating the average of the reported outcomes of the treated and untreated groups, since ignorability guarantees the comparability of the average results between the two groups without any further adjustment.

- L 53. Shouldn’t it be “(…) Y1 and Y0 are the potential outcomes (…)”?

Authors: We have corrected the sentence.

- L. 55. “This does not apply to observational studies.” Disagree, it does apply also in observational studies, as this is just the definition of the effect of interest. However, you cannot simply take the means of both groups to estimate both effects, so the way to identify and estimate the estimand is different, but the estimand can still be defined as the ATE.

Authors: We agree that the sentence was unclear. We have rephrased it to clarify that the ATE can also be estimated using observational studies.

- L 57. “preferred” compared to what? Furthermore, alternative modeling approaches, like IPW-estimation, can also "(…) quantify effects related to the dependent variable while controlling for a limited set of covariates.", so why is this characteristic of regression models an advantage per se?

Authors: We have modified the sentence dropping the comparative verb. 

- L. 61. Not true, as the presence of multiple confounders is not necessarily a problem for outcome regression methods. Furthermore, model misspecification can also occur in IPW methods (when your exposure model is misspecified), so I don’t understand the point being made here.

Authors: The sentence has been modified considering other issues in covariate adjustment as collinearity, over bias adjustment, and data sparsity. 

- L. 68. Does the target audience of this paper know what is meant by "overlapping distribution of covariates"?

Authors: We have rephrased the sentence to make the concept clearer to a broader audience.

- L. 70-76. First, the authors introduce propensity scores. Then, in lines 70 to 76, the authors talk about stratification based on individual covariates. This seems like a weird shift? Wouldn’t you stratify based on the propensity score? Furthermore, in the lines 70 to 73, it is mentioned that we need to match for confounders, and then in line 74 it is said that we need to additionally consider “(…) characteristics that influence both the probability of the exposure and outcome (…)”. How are these characteristics in line 74 different from the confounders in lines 70 to 73?

Authors: We have revised the first sentence (70-73) and moved the second sentence (74-76), which explains the importance of confounders, to an earlier position for clarity.

- Line 76-79. The sentence is unclear to me. What is also unclear is how reference 16 is relevant, which is about extending results from an RCT to a wider target population of interest, to stratification that is discussed earlier in this paragraph.

Authors: In the new version of the introduction these lines have been removed. 

- Line 81. Although IPW estimation is the main focus of the paper, it is only mentioned in line 81. This is confusing as matching is discussed earlier in the introduction, but actually not at all what the paper focusses on.

Authors: We have now introduced the IPTW concept earlier in the Introduction section for better clarity.

- L. 110. What is a “likely” impact?

Authors: We agree that the word "likely" can be unclear. We have removed it from the sentence.

- L. 112. Confounders are not just “thought to be responsible”, the ARE actually responsible for difference in characteristics between treated and untreated. Furthermore, should a confounder also impact the outcome, for it to be a confounder? This is missing from this sentence.

Authors: This sentence has been deleted. We feel that the meaning of a confounder is now well explained in the Introduction section.

- L. 114. What does “specification of several assumptions to be correctly identified” mean?

Authors: We have combined this sentence with the second one to improve readability.

- L. 116. Why is “no interaction with the treatment variable” a key assumption? As long as I include an interaction when there is one, then I can still estimate my ATE. Admittedly, the coefficient itself is not the ATE anymore, but you can still estimate the ATE with g-computation, which involves multiple regression analysis.

Authors: We agree with the comment and we have removed “interaction with the treatment variable” from the list. 

- L. 136. Here three fundamental assumptions are discussed, but aren’t these the assumptions of exchangeability and positivity, discussed 1 sentence earlier?

Authors: We dropped the first two items on the list, which are a consequence of the exchangeability and positivity assumptions defined in the previous sentence. We kept model (mis)specification, the third item on the list.

- Figure 2. It needs further explanation, because I have no idea what I am looking at.

Authors: We have expanded this paragraph to better describe the purpose of the figure and the descriptive results derived from it.

- L. 217. The statement that GBM provided more precises estimates seems unnuanced. Perhaps for the scenario with a quadratic effect of Z, but not for the scenario with only a linear effect. Also, are you worried about overfitting when using GBMs? Is that an issue?

Authors: We have corrected the sentence to emphasize the superiority of flexible methods like GBM when the relationship between Z and X is not linear. The problem of overfitting is a real issue for GBM, and there are several control parameters that can help mitigate this effect. However, we adopt the default arguments of the method_gbm function of the R package WeightIt. We have made some attempts and for our simulation condition the suggested setting works well.

- L. 228. Given that IPTW-based method can be quite inefficient, it seems appropriate to also look at a measure of variability of the estimates. For example, the MSE?

Authors: We have now calculated and reported the MSE for both the scenarios.

- L. 231. I do not see how this statement is backed up by the results. Comparing n = 200 to n = 500, relative bias is lower for n = 500. However, then it increases again. Furthermore, what is the uncertainty around these ARebias estimates? I think they overlap, as the point estimates of ARebias are so close. In that case, you wouldn’t conclude that ARebias decreased with increasing sample size.

Authors: We agree with the comment. In the light of new results, we have changed the sentence according to the uncertainty of the standard deviation of ARebia.

- L. 234. Why is there a heightened risk of bias in all cases, as all relationships have been correctly specified, right? (i.e., they are all linear)

Authors: In the new simulation, this phenomenon disappears, especially for large sample sizes. For small sample sizes, it can be attributed to the limited information for adjusting the estimates and implies a positive bias as reported for method A (

---

## [Decision Letter · Decision Letter 2]

24 Sep 2024

PONE-D-24-03437R2Impact of COVID-19 Vaccination on Preventive Behavior: The Importance of Confounder Adjustment in Observational StudiesPLOS ONE

Dear Dr. Caserotti,

Thank you for submitting your manuscript to PLOS ONE. After careful consideration, we feel that it has merit but does not fully meet PLOS ONE’s publication criteria as it currently stands. Therefore, we invite you to submit a revised version of the manuscript that addresses the points raised during the review process.

In the next revision, please address the comments made by the new reviewer and consider incorporating the literature they mentioned.

We look forward to receiving your revised manuscript.

Kind regards,

Srebrenka Letina, Ph.D.

Academic Editor

PLOS ONE

Journal Requirements:

Reviewers' comments:

Reviewer's Responses to Questions

**Comments to the Author**

1. If the authors have adequately addressed your comments raised in a previous round of review and you feel that this manuscript is now acceptable for publication, you may indicate that here to bypass the “Comments to the Author” section, enter your conflict of interest statement in the “Confidential to Editor” section, and submit your "Accept" recommendation.

Reviewer #3: (No Response)

2. Is the manuscript technically sound, and do the data support the conclusions?

Reviewer #3: Partly

3. Has the statistical analysis been performed appropriately and rigorously? 

Reviewer #3: N/A

4. Have the authors made all data underlying the findings in their manuscript fully available?

Reviewer #3: (No Response)

5. Is the manuscript presented in an intelligible fashion and written in standard English?

Reviewer #3: Yes

6. Review Comments to the Author

Reviewer #3: Report on

PONE-D-24-03437R2

Impact of COVID-19 Vaccination on Preventive Behavior: The Importance of Confounder Adjustment in Observational Studies

The paper investigates effect of vaccination on preventive behaviors by controlling estimation biases through appropriate method. Further, simulations are conducted to evaluate the degree of biases under various settings. I believe, the results are convincing and valuable for understanding interaction relation between vaccination and preventive behaviors. However, basic information about PSM is not sufficiently provided. This study would contribute to the literature if some additional works have been appropriately done. Specific comments are as follows.

1. I would like to see the results in the first stage of Propensity score matching. That is, present the results for estimating propensity score (in appendix).

2. How did authors set the caliper to Author should conduct estimations using pooled sample and then put the cross terms between gender dummy (for instance male dummy when reference group is female) and key variables. Cite the following paper and show how to set caliper.

Wang Y, Cai H, Li C, Jiang Z, Wang L, Song J, et al. (2013) Optimal Caliper Width for Propensity Score Matching of Three Treatment Groups: A Monte Carlo Study. PLoS ONE 8(12): e81045. https://doi.org/10.1371/journal.pone.0081045

3. Indicates the effective sample sizes by illustrating Figures. Authors should cite Endo (2022) and follow the style of Figure 4 of Endo(2022).

Endo A. ‘Not finding causal effect’ is not ‘finding no causal effect’ of school closure on COVID-19. F1000Research 2022, 11:456

https://doi.org/10.12688/f1000research.111915.1

4. The following works are closely related to the paper. From the theoretical viewpoint to consider moral hazard, Peltzman effects (1975) should be noted. Considering the incentive to engage in preventive behaviors against COVID 19, authors should cite the following papers.

Asri A, Asri V, Renerte B, Föllmi-Heusi F, Leuppi JD, Muser J, et al. Wearing a mask-For yourself or for others? Behavioral correlates of mask wearing among COVID-19 frontline workers. PLOS ONE. 2021;16 (7 July):e0253621. doi:10.1371/journal.pone.0253621.

Cato S, Iida T, Ishida K, Ito A, Katsumata H, McElwain KM, et al. Vaccination and altruism under the COVID-19 pandemic. Public Health Pract (Oxf). 2022;3:100225. doi:10.1016/j.puhip.2022.100225.

Cato S, Iida T, Ishida K, Ito A, McElwain KM, Shoji M. Social distancing as a public good under the COVID-19 pandemic. Public Health. 2020;188:51-3. doi:10.1016/j.puhe.2020.08.005.

Hossain MdE, Islam MdS, Rana MdJ, Amin MR, Rokonuzzaman M, Chakrobortty S, et al. Scaling the changes in lifestyle, attitude, and behavioral patterns among COVID-19 vaccinated people: Insights from Bangladesh. Hum Vaccin Immunother. 2022;18:2022920. doi:10.1080/21645515.2021.2022920.

Peltzman S. The effects of automobile safety regulation. J Pol Econ. 1975;83:677-725. doi:10.1086/260352.

Yamamura, E.　Tsutsui, Y., Kosaka, Y., Ohtake, F. 2023. Association between the COVID-19 vaccine and preventive behaviors: Panel data analysis from Japan. Vaccines, 11(4), 810; https://doi.org/10.3390/vaccines11040810

Yamamura, E.　Tsutsui, Y., Ohtake, F. 2024. The COVID-19 vaccination, preventive behaviours and pro-social motivation: Panel data analysis from Japan. 　 Humanities and Social Sciences Communications, 11, Article number: 476

5. In Tables 1 and 2, I cannot find information about what number in parentheses for “ARebias” in columns A, B, C, and D. To take an example, in the first line, “A” shows 1.318 (1.032). Is “(1.032)” standard error? Authors should explain it in the footnote of Tables 1 and 2.

6. In the PDF file of the manuscript, there seems to be three Figures as Figure 2 because there are three pages. Puzzlingly I cannot find two of Figures 2 because of blank on two pages although I find the first Figure 2 (Bars of “A” and “B”). Further, Figure 1 and the first of Figure 2 are too simple to be meaningful. Probably, these are more sophisticated with more useful information. I do not know the reason of poor presentation and disappearance of these Figures. Please made PDF file again to appropriately demonstrate these Figures. After checking it, I would like to provide final evaluation.

7. PLOS authors have the option to publish the peer review history of their article (what does this mean?). If published, this will include your full peer review and any attached files.

Reviewer #3: **Yes: **Eiji YAMAMURA

---

## [Author Response · Author response to Decision Letter 2]

8 Oct 2024

Editor

Thank you for submitting your manuscript to PLOS ONE. After careful consideration, we feel that it has merit but does not fully meet PLOS ONE’s publication criteria as it currently stands. Therefore, we invite you to submit a revised version of the manuscript that addresses the points raised during the review process.

In the next revision, please address the comments made by the new reviewer and consider incorporating the literature they mentioned.

Authors: Thank you for the possibility to revise the manuscript. In the current version of the paper we have replied to all the reviewer’ comments, incorporating the suggested literature where appropriate. 

Reviewer 3: Review for PLOS ONE, manuscript PONE-D-24-03437_R2

Title: Impact of COVID-19 Vaccination on Preventive Behavior: The Importance of Confounder Adjustment in Observational Studies

The paper investigates effect of vaccination on preventive behaviors by controlling estimation biases through appropriate method. Further, simulations are conducted to evaluate the degree of biases under various settings. I believe, the results are convincing and valuable for understanding interaction relation between vaccination and preventive behaviors. However, basic information about PSM is not sufficiently provided. This study would contribute to the literature if some additional works have been appropriately done. Specific comments are as follows.

1. I would like to see the results in the first stage of Propensity score matching. That is, present the results for estimating propensity score (in appendix).

Authors: The results of the first stage of propensity score weighting have been moved from the supplementary material to the results section (new Figure 3).

2. How did authors set the caliper to Author should conduct estimations using pooled sample and then put the cross terms between gender dummy (for instance male dummy when reference group is female) and key variables. Cite the following paper and show how to set caliper.

Wang Y, Cai H, Li C, Jiang Z, Wang L, Song J, et al. (2013) Optimal Caliper Width for Propensity Score Matching of Three Treatment Groups: A Monte Carlo Study. PLoS ONE 8(12): e81045. https://doi.org/10.1371/journal.pone.0081045

Authors: As we used a propensity score weighting procedure, we had not considered a caliper stage, as all observations are weighted. In practice, if we estimate a weight close to zero, we do not include that observation in the analysis. To the best of our knowledge, the use of caliper is mainly related to propensity score matching to avoid controls (or cases) that have a low degree of matching with cases (or controls) after matching. Regarding the second part of the comment, as suggested, we included the interaction terms between gender and three key variables (age class, educational level, COVID-19 risk perception and conspiracy score) in both the propensity score model and the ordinal cumulative regressions.

3. Indicates the effective sample sizes by illustrating Figures. Authors should cite Endo (2022) and follow the style of Figure 4 of Endo (2022).

Endo A. ‘Not finding causal effect’ is not ‘finding no causal effect’ of school closure on COVID-19. F1000Research 2022, 11:456 https://doi.org/10.12688/f1000research.111915.1

Authors: We have added the effective sample size in the new Figure 3 (ex Figure S1). 

4. The following works are closely related to the paper. From the theoretical viewpoint to consider moral hazard, Peltzman effects (1975) should be noted. Considering the incentive to engage in preventive behaviors against COVID 19, authors should cite the following papers.

Asri A, Asri V, Renerte B, Föllmi-Heusi F, Leuppi JD, Muser J, et al. Wearing a mask-For yourself or for others? Behavioral correlates of mask wearing among COVID-19 frontline workers. PLOS ONE. 2021;16 (7 July):e0253621. doi:10.1371/journal.pone.0253621.

Cato S, Iida T, Ishida K, Ito A, Katsumata H, McElwain KM, et al. Vaccination and altruism under the COVID-19 pandemic. Public Health Pract (Oxf). 2022;3:100225. doi:10.1016/j.puhip.2022.100225.

Cato S, Iida T, Ishida K, Ito A, McElwain KM, Shoji M. Social distancing as a public good under the COVID-19 pandemic. Public Health. 2020;188:51-3. doi:10.1016/j.puhe.2020.08.005.

Hossain MdE, Islam MdS, Rana MdJ, Amin MR, Rokonuzzaman M, Chakrobortty S, et al. Scaling the changes in lifestyle, attitude, and behavioral patterns among COVID-19 vaccinated people: Insights from Bangladesh. Hum Vaccin Immunother. 2022;18:2022920. doi:10.1080/21645515.2021.2022920.

Peltzman S. The effects of automobile safety regulation. J Pol Econ. 1975;83:677-725. doi:10.1086/260352.

Yamamura, E.　Tsutsui, Y., Kosaka, Y., Ohtake, F. 2023. Association between the COVID-19 vaccine and preventive behaviors: Panel data analysis from Japan. Vaccines, 11(4), 810; https://doi.org/10.3390/vaccines11040810

Yamamura, E.　Tsutsui, Y., Ohtake, F. 2024. The COVID-19 vaccination, preventive behaviours and pro-social motivation: Panel data analysis from Japan. 　 Humanities and Social Sciences Communications, 11, Article number: 476

Authors: We thank the reviewer for his suggestions; we have now considered the theoretical viewpoint of moral hazard in explaining behavioral effect of COVID-19 vaccination, including some of the suggested references in the discussion section

5. In Tables 1 and 2, I cannot find information about what number in parentheses for “ARebias” in columns A, B, C, and D. To take an example, in the first line, “A” shows 1.318 (1.032). Is “(1.032)” standard error? Authors should explain it in the footnote of Tables 1 and 2.

Authors: We have included in the headings of Table 1 and 2 specifying that the value in parentheses is the standard error of the estimates. 

6. In the PDF file of the manuscript, there seems to be three Figures as Figure 2 because there are three pages. Puzzlingly I cannot find two of Figures 2 because of blank on two pages although I find the first Figure 2 (Bars of “A” and “B”). Further, Figure 1 and the first of Figure 2 are too simple to be meaningful. Probably, these are more sophisticated with more useful information. I do not know the reason of poor presentation and disappearance of these Figures. Please made PDF file again to appropriately demonstrate these Figures. After checking it, I would like to provide final evaluation.

Authors: We are very sorry for this inconvenience. We have double-checked the figure files that are in PDF format. All figures are correctly reported in the proof version of the submission and we hope that the problem has now been resolved.

---

## [Decision Letter · Decision Letter 3]

18 Oct 2024

Impact of COVID-19 Vaccination on Preventive Behavior: The Importance of Confounder Adjustment in Observational Studies

PONE-D-24-03437R3

Dear Dr. Caserotti,

We’re pleased to inform you that your manuscript has been judged scientifically suitable for publication and will be formally accepted for publication once it meets all outstanding technical requirements.

Kind regards,

Srebrenka Letina, Ph.D.

Academic Editor

PLOS ONE

Additional Editor Comments (optional):

Reviewers' comments:

Reviewer's Responses to Questions

**Comments to the Author**

1. If the authors have adequately addressed your comments raised in a previous round of review and you feel that this manuscript is now acceptable for publication, you may indicate that here to bypass the “Comments to the Author” section, enter your conflict of interest statement in the “Confidential to Editor” section, and submit your "Accept" recommendation.

Reviewer #3: All comments have been addressed

2. Is the manuscript technically sound, and do the data support the conclusions?

Reviewer #3: Yes

3. Has the statistical analysis been performed appropriately and rigorously? 

Reviewer #3: Yes

4. Have the authors made all data underlying the findings in their manuscript fully available?

Reviewer #3: Yes

5. Is the manuscript presented in an intelligible fashion and written in standard English?

Reviewer #3: Yes

6. Review Comments to the Author

Reviewer #3: Authors have properly improved the paper to follow all of my commments. I believe that the revised version of paper will meet the criteria for publicaiton. Congratulation!

7. PLOS authors have the option to publish the peer review history of their article (what does this mean?). If published, this will include your full peer review and any attached files.

Reviewer #3: **Yes: **Eiji YAMAMURA

---

## [Editor Report · Acceptance letter]

31 Oct 2024

PONE-D-24-03437R3 

PLOS ONE

Dear Dr. Caserotti, 

I'm pleased to inform you that your manuscript has been deemed suitable for publication in PLOS ONE. Congratulations! Your manuscript is now being handed over to our production team.

Kind regards, 

on behalf of

Dr. Srebrenka Letina 

Academic Editor

PLOS ONE